

# A UAV-based active AirCore system for accurate measurements of greenhouse gases

Truls Andersen[1], Bert Scheeren[1], Wouter Peters[1, 2], Huilin Chen[1, 3]

[1]Centre for Isotope Research (CIO), Energy and Sustainability Research Institute Groningen (ESRIG), University of
Groningen, the Netherlands
[2]Meteorology and Air Quality, Wageningen University and Research Center, the Netherlands
[3]Cooperative Institute for Research in Environmental Sciences (CIRES), University of Colorado, Boulder, Colorado, USA

*Correspondence to*: Huilin Chen (Huilin.Chen@rug.nl)

**Abstract.** We developed and field-tested a UAV-based active AirCore for atmospheric mole fraction measurements of $CO_2$, $CH_4$, and CO. The system applies an alternative way of using the AirCore technique invented by NOAA. As opposed to the conventional concept of passively sampling air using the atmospheric pressure gradient during descent, the active AirCore collects atmospheric air samples using a pump to pull the air through the tube during flight, which opens up the possibility to sample atmospheric air in both vertical and horizontal planes. The active AirCore used for this study weighs ~1.1 kg. It consists of a ~50 m long 1/8" stainless steel tube with a 0.005" wall thickness, a 7.5 cm 1/4" stainless steel tube filled with magnesium perchlorate, a small KNF micropump and a 45 μm orifice working together to form a critical flow of dried atmospheric air through the active AirCore. A cavity ring-down spectrometer (CRDS) was used to analyze the air samples on site shortly after landing for mole fraction measurements of $CO_2$, $CH_4$, CO, and $H_2O$.

We flew the active AirCore system on a UAV near the atmospheric measurement station at Lutjewad, located in the northwest of the city of Groningen in the Netherlands. Five consecutive flights took place over a five-hour period in the same morning, starting at sunrise until noon. We validated the measurements of $CO_2$ and $CH_4$ from the active AirCore against those from the Lutjewad station at 60 m. The results show a good agreement between the measurements from the active AirCore and the atmospheric station (N = 146, $R^2_{CO2}$: 0.97 and $R^2_{CH4}$: 0.94, and mean differences: $\Delta_{CO2}$: 0.18 ppm; $\Delta_{CH4}$: 5.13 ppb). The vertical and horizontal resolution at typical UAV speeds of 1.5 m/s and 2.5 m/s were determined to be ± 26.4 to 28.2 m and ± 44.0 to 47.0 m respectively, depending on the storage time. The collapse of the nocturnal boundary layer and the build-up of the mixed layer were clearly observed with three consecutive vertical profile measurements in the early morning hours. Besides this, we furthermore detected a $CH_4$ hotspot in the coastal wetlands from a horizontal flight north to the dike, which demonstrates the potential of this new active AirCore method to measure at locations where other platforms have no practical access.



# 1 Introduction

Since the 18th century industrial revolution, greenhouse gas (GHG) mole fractions have been increasing due to anthropogenic activity. Rapid increases in carbon dioxide ($CO_2$) and methane ($CH_4$) have occurred since the 1950's, contributing to global climate change (IPCC, 2014a; IPCC, 2014b). Understanding and quantifying both natural and

anthropogenic fluxes of the two major GHGs, namely $CO_2$ and $CH_4$, is vital to predict future mole fraction levels, and to help monitor the effectiveness of the emissions reduction efforts.

Both $CO_2$ and $CH_4$ are naturally occurring greenhouses gases in our atmosphere, with $CO_2$ the more abundant of the two. Today, natural production of $CO_2$ happens mainly through decay of organic matter and respiration by aerobic organisms. Besides the natural sources of atmospheric $CO_2$, there are additional anthropogenic contributions to the total atmospheric

$CO_2$ mole fractions, mainly from burning of fossil fuels. In recent years the mole fractions of atmospheric $CO_2$ have been increasing by ~ 2 ppm (parts per million) per year (Tans and Keeling, 2017; Hartmann et al., 2013). Methane has a shorter lifetime in the atmosphere compared to that of $CO_2$, but $CH_4$ is more efficient at trapping radiation. The comparative impact of $CH_4$ on climate change is 20-30 times greater than that of $CO_2$ over a 100-year period (Saunois et al., 2016; Dlugokencky et al., 2011). Methane is naturally produced and emitted to the atmosphere when organic matter decomposes in low oxygen

environments, and natural sources include wetlands, swamps, marshes, termites, and oceans. From 2007 to 2016, the increase of the global methane mole fractions has been ~ 7 ppb (parts per billion) per year (Hartmann et al., 2013). The main contributors to anthropogenic methane emissions are leakages from coal mining and oil the oil and gas industry, ruminant animals, rice agriculture, waste management, and biomass burning (Kirschke et al., 2013; Saunois et al., 2016). The quantification of $CH_4$ emissions is highly important in studying the global methane cycle where vertical profiling with high

resolution provide further information on the contributions from $CH_4$ sources and sinks (Berman et al., 2012).

Presently, greenhouse gases like $CO_2$ and $CH_4$ are monitored via a network of global ground-based atmospheric monitoring sites. These ground-based monitoring sites provide stationary measurements of greenhouse gases at the earth's surface (Hartmann et al., 2013). Although essential to infer surface fluxes, these ground-based monitoring stations lack information about the vertical distribution of the atmospheric mole fractions. Several satellite-based missions monitoring greenhouse

gases from space have since been developed to improve spatial coverage and monitoring of atmospheric trace gases. Short-wave infrared (SWIR) satellites can observe and retrieve information about the total atmospheric column, mainly during daytime and on land. Several SWIR missions have run in the past decade. The Scanning Imaging Absorption Spectrometer for Atmospheric Cartography (SCIAMACHY) started in 2003 and was discontinued in 2012 (Frankenberg et al., 2011; Butz et al., 2011; Wecht et al., 2014). The Greenhouse Gases Observing Satellite (GOSAT) has been operational since 2009 and

still performs global coverage trace gas measurements to date (Butz et al., 2011). The most recent one, the Orbiting Carbon Observatory 2 (OCO-2), was launched in 2014, and only monitors radiances to retrieve $CO_2$ columns (Nelson et al., 2016). Observations of terrestrial radiation in the thermal infrared (TIR) provide worldwide, 24-hour information about the mid-tropospheric columns, and several missions have been operational since 2002. Missions include the Atmospheric Infrared



Sounder (AIRS) which has been operational since 2002 (Crevoisier et al., 2003; Xiong et al., 2010), the Tropospheric Emission Spectrometer (TES) which started in 2004 and was discontinued in 2011 (Worden et al., 2012), and the Infrared Atmospheric Sounding Interferometer (IASI) which has been operational since 2007 (Crevoisier et al., 2009a; Crevoisier et al., 2009b; Xiong et al., 2013). These satellite-based vertical profiles mainly cover the upper troposphere and lower stratosphere with a low vertical resolution (Foucher et al., 2011).

Satellite-based sounding systems help to bring a better understanding of greenhouse gas distribution throughout different layers of the atmosphere, and has the advantages of global coverage and multi species detection. However, uncertainties are high and require complimentary in-situ measurements with higher accuracy, response time, and spatial resolution to reduce uncertainties in the overall atmospheric columns (Jacob et al., 2010). Highly accurate vertical profiling is required to study large carbon sources and sinks, where satellite data is insufficient for current climate modeling efforts (Hartmann et al., 2013). The in-situ measurements also provide an additional verification to the satellite observations (Berman et al., 2012).

Throughout the years, several aircraft missions have contributed with regular measurements along commercial airlines to provide additional vertical information, such as the CONTRAIL project (Machida et al., 2008) and the CARIBIC project (Schuck et al., 2009). Less regular aircraft campaigns are also dedicated to study GHG at a more local scale (Chen et al., 2010; Karion et al., 2013; Zhang et al., 2014; Sweeney et al., 2015), or from pole-to-pole, such as the HIPPO project (Wofsy, 2011). These vertical profiles are usually limited to 150 m - 14 km.

The limitations met by aircraft missions lead to new developments in instrumentation for measuring $CO_2$ and $CH_4$. This includes the Fourier Transform Spectrometer (FTS) measurements within the TCCON network (Wunch et al., 2010), cryogenic discrete flask sample measurements in the stratosphere using high-altitude balloons (Engel et al., 2009), and laser diode spectrometers such as the Pico-SDLA instruments (Durry et al., 2004; Ghysels et al., 2011; Joly et al., 2007). These systems are heavy and require massive balloon-borne platforms, which have proven difficult to fly on a regular basis.

In 2010, the National Oceanic and atmospheric Administration (NOAA) developed the first AirCore, an innovative atmospheric air sampling system (Karion et al., 2010) from an idea originally developed and patented by Pieter Tans (Tans, 2009). The AirCore consists of long, thin-wall stainless steel tubing capable of sampling and preserving atmospheric profile information. The AirCore is evacuated as it is lifted up to a high altitude ($\sim$ 30 km) by a balloon, and then during descent after the balloon bursts, it is passively filled with atmospheric air samples due to the increasing ambient pressure. The samples are analyzed on the ground to retrieve the GHG vertical profiles. The length and diameter of the tubes, and the time it takes from sampling until analysis ultimately determine the vertical resolution. Since the first development of the AirCore (Karion et al., 2010), additional augmentations of the AirCore has been developed and tested. This includes smaller and lighter AirCores developed at Goethe University Frankfurt (Engel et al., 2017), University of Groningen (Paul et al., 2016; Chen et al., 2017) and a high-resolution (HR) AirCore developed at Ecole polytechnique, Université Paris-Saclay (Membrive et al., 2017). Other applications using the AirCore technique includes measurements of $\delta^{13}CH_4$ and $C_2H_6/CH_4$ ratios, using the AirCore to store a rapidly acquired sample and analyze the sample at a lower flow rate while maintaining the sample integrity (Rella et al., 2013).





In recent years, the use of unmanned aerial vehicles (UAVs) has become a new complimentary platform for GHG measurements. Previous studies include the atmospheric monitoring of point source fossil fuel $CO_2$ and $CH_4$ from a gas treatment plant using a Helikite (Turnbull et al., 2014), $CO_2$, $CH_4$ and $H_2O$ measurements onboard the National Aeronautics and Space Administration (NASA) Sensor Integrated Environmental Remote Research Aircraft (SIERRA) UAV (Berman et

al., 2012) and the quantification of $CH_4$ mole fractions and isotopic compositions from heights up to 2700 m on Ascension Island using a remotely piloted octo copter (Lowry et al., 2015; Brownlow et al., 2016).

For this study, we combine the flexibility and mobility of UAVs, and the AirCore's ability to capture and preserve the spatial resolution of atmospheric air samples to design and develop an alternative AirCore version, named active AirCore. Instead of passively sampling air due to the changing ambient pressure during flight, the active AirCore pulls atmospheric air

samples through the tube at a certain flow rate using a small micropump. This allows for a highly mobile system that can obtain both vertical and horizontal profiles with a high spatial resolution. Unlike the original AirCore (Karion et al., 2010) and the newer versions (Membrive et al., 2017; Engel et al., 2017; Chen et al., 2017) that are all made to sample the atmospheric column including the stratosphere, the active AirCore has been designed to fulfill a different purpose, and does not aim to reach a height well above the troposphere like its predecessors. With the capability of sampling horizontal

transects, the active AirCore can help quantify $CO_2$ and $CH_4$ emissions from local areas such as wetlands, landfills and other $CH_4$ hot spots, and quantify point sources emissions from such as power plant plumes. It can also provide highly accurate and precise measurements to be used for validation of measurements of remote sensing techniques.

The instrument design is presented in the method section together with the experimental setup and the data processing

method. The results section presents the measurements made by the active AirCore during five flights in a day. Section 4 discusses the horizontal and vertical resolution. Section 5 presents the conclusions.

## 2 Method

The active AirCore, designed to fly with a lightweight UAV, consists of ~50 m thin-wall stainless steel tubing, a dryer, a micropump, and a datalogger. It is placed in a carbon fiber box and attached to the UAV using two carbon fiber rods. Prior

to every flight, the active AirCore is flushed with a calibrated fill gas that is spiked with ~ 10 ppm CO, which helps to identify the starting point of ambient air sampling during later analysis. The active AirCore starts to collect air samples when the micropump is turned on using a switch located outside the box shortly before a UAV flight, and the pump is turned off after the UAV lands. Air samples are collected during the flight and retained within the active AirCore. The active AirCore samples are then immediately analyzed with a trace gas analyzer.

## 2.1 Active AirCore

The dimensions of the active AirCore, along with some key parameters, are given in table 1.



**Table 1: the dimensions and key parameters of the active AirCore.**

| | |
|---|---|
| Length | 49.1 ± 0.1 m |
| Tubing | 304-grade stainless steel |
| Outer Diameter (OD) | 3.175 ± 0.051 mm (1/8 ± 0.002 in.) |
| Wall thickness | 0.127 ± 0.051 mm (0.005 ± 0.002 in.) |
| Coating | SilcoNert 1000, by Restek Inc. |
| AirCore tubing weight | 431.2 ± 0.1 g |
| AirCore volume | 358 ± 10 ml |
| Total payload weight | 1131 ± 1 g |
| Vertical spatial resolution $CO_2$/$CH_4$ (1.5 m/s) | 26.4 to 28.2 m |
| Horizontal spatial resolution $CO_2$/$CH_4$ (2.5 m/s) | 44.0 to 47.0 m |

As the thin-wall tubing is very fragile, we have used custom-made stainless-steel connectors to reinforce the connection with the coiled tube and Swagelok fittings at both ends. These connectors have an ID (inner diameter) of 3.275 mm on one end,

and an OD (outer diameter) of 3.175 mm on the other. The 3.175 mm ID of the connector is inserted onto the thin-walled AirCore tubing, and fastened using ceramic glue, leaving the 3.175 mm OD side open and usable by Swagelok fittings. To obtain a constant flow through the AirCore, an orifice (OD ¼ in., orifice diameter 45 ± 10% μm, Lenox laser Inc.) is placed between the pump and the coiled tube. The upstream pressure of the orifice is close to ambient, or more accurately the ambient pressure minus a small pressure drop across the whole coiled tube, while the downstream pressure of the orifice is

mainly determined by the pumping capacity and was measured at 380 hPa with the pump (KNF micropump, model 020L) in the laboratory. Thus, the flow across the coiled tube is expected to be critical as long as the upstream pressure is above ~760 hPa (2 x 380 hPa), or below ~ 2.4 km above the sea level, and was measured to be 21.5 sccm (standard cubic centimeter per minute) in the laboratory. The pressure between the orifice and the pump is constantly monitored through a stainless-steel Swagelok tee junction. The pump and the tee junction are connected via flexible fluorinated ethylene propylene (Tygon)

tubing (1/8 in. ID). This same type of tubing is also connected to the outlet of the pump and leads to a hole on the side of the box, venting the pump exhaust outside of the box. Air samples are dried with a 7.5 cm long stainless-steel tube (1/4 in. OD) filled with magnesium perchlorate before they are sampled into the coiled tube. The inlet of the active AirCore system is placed at the bottom of the carbon fiber box, and is attached through a hole to the dryer tube with a small piece of flexible ¼ in. ID nylon tubing.

The box itself is made from 0.5 mm thick carbon fiber plate with a density of 1600 kg/m³, providing a sturdy and lightweight box to house the active AirCore system. The box has a length of 34 cm, a width of 19.5 cm and a height of 12





cm. The total weight of the active AirCore system with the box is 1.1 kg. Figure 1a shows a schematic design of the active AirCore system, while figure 1b shows a photo of the prototype product.

The datalogger is made using an Arduino MEGA 2650 board that records meteorological data via two pressure sensors, five temperature sensors, a relative humidity sensor, and a GPS (Global Positioning System) receiver. The pressure sensors are silicon pressure sensors of the model Honeywell TruStability HSC. One pressure sensor monitors the pressure between the pump and the orifice, while the other measures the outside ambient pressure through a flexible nylon tube going through the bottom of the box. These sensors have an accuracy of ± 0.25% in the range of 67 - 1034 hPa (1 − 15 psi). The relative humidity is a model DHT22, which measures in a range from 0 − 100% with an uncertainty of 2.5%. The temperature sensor

embedded in the relative humidity sensor can measure in a range of - 40 to 125 °C with an uncertainty of 0.5 °C. During the day of this study, we did however not have relative humidity measurements, due to the sensor being placed inside the enclosed AirCore box. The external temperature sensors are all PT100 elements from Innovative Sensor Technology, and have an uncertainty of 0.15 °C. The GPS coordinates and time are measured using a GPS model ATM2.5 NEO-6M module with EEPROM built-in activity.

The datalogger is powered by one 9 V battery, while the micro pump is powered by 12 V, four 3 V batteries connected in series. The micro pump was controlled via an on/off switch mounted on the outside of the carbon fiber fox for easy use before take-off.

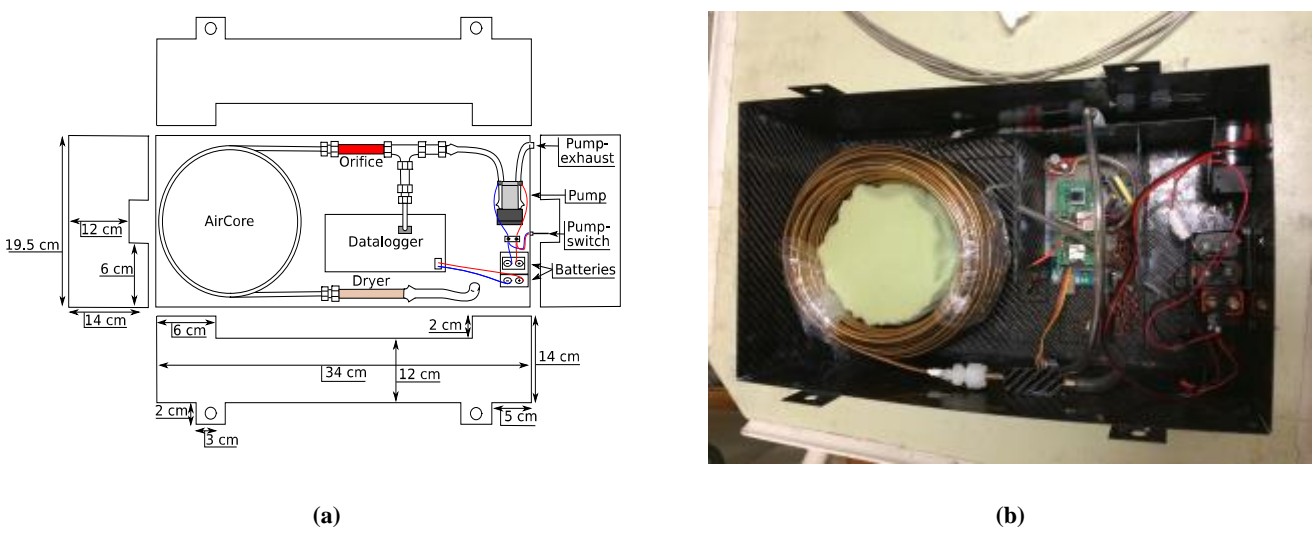

(a)                                                                      (b)

**Figure 1: (a) a schematic design of the UAV AirCore system. (b) an image of the UAV AirCore system**

**2.2 The trace gas analyzer**

All mole fraction analyzes of air samples from the active AirCore are conducted using a cavity ring-down spectrometer (CRDS, Picarro, Inc., CA, model G2401) (Crosson, 2008). This particular model can measure $CO_2$, $CH_4$, CO, and $H_2O$ with



high precision. The cavity of the analyzer is strictly maintained at a pressure of ~ 186 hPa (140 Torr) and a temperature of 45 °C to achieve a precision (one-sigma, 0.5 Hz) better than 0.05 ppm for $CO_2$, 1 ppb for $CH_4$, and 10 ppb for CO. We control the sample flow of the analyzer operating in the inlet valve control mode at a constant rate using a needle valve between the analyzer and the vacuum pump. We set the flow rate during all the analyses of active AirCore samples at ~ 20.5

sccm. After each analysis, the analyzer is switched to measure fill gas through the active AirCore at a higher flow rate of ~ 120 sccm by fully opening the needle valve. In this way, we are able to shorten the time interval between one to the next flight to 50 minutes.

**2.3 Laboratory tests**

Prior to the flights, we validated the active AirCore measurements in laboratory experiments against in situ mole fraction

measurements of $CO_2$, $CH_4$, and CO using a CRDS analyzer. During the experiments, the CRDS analyzer and the active AirCore were set up to sample the roof air through the same inlet via a tee junction. The roof air was partially dried, having a water vapor content of ~ 0.1%. The water vapor effects were corrected based on Chen et al., (2010) and Rella et al. (2013) for $CO_2$ and $CH_4$, and Chen et al. (2013) for CO to obtain dry mole fractions of $CO_2$, $CH_4$, and CO, respectively. Both the analyzer and the AirCore were flushed with dry cylinder air prior to the start of the test, until the measured water vapor level

was below 0.005%. Once the active AirCore was fully sampled, the micro pump was turned off and a shut-off valve was switched to close the inlet. This was followed by the analysis of the active AirCore samples using the same CRDS analyzer. A three-way valve at the end of the active AirCore was also turned so that the sample was chased by dry cylinder air with known mole fractions. The flow rate through the CRDS analyzer during analysis was 19.2 sccm, while the air samples were collected into the active AirCore at a flow rate of 21.5 sccm. Once the test was complete, the time stamp of the active

AirCore data was compressed using the ratio of the sampling flow rate to the analysis flow rate, i.e. 21.5/19.2. The active AirCore data was then shifted back in time to overlay with the direct roof air analysis. Several experiments were performed to verify the consistency of the results, and we observed a strong correlation between the direct CRDS analyzer measurements and the sampled active AirCore mole fraction values. The $R^2$ values were 0.99, 0.97, and 0.97, with the mean differences of 0.04 ± 0.21 ppm, 0.58 ± 0.67 ppb and 0.86 ± 27.37 ppb for $CO_2$, $CH_4$, and CO, respectively. The large

standard deviation in CO is due to a sharp spike of several hundred ppb during three experiments.

**2.4 The UAV**

The active AirCore system has been flown aboard a small quad copter UAV (model DJI Inspire 1 Pro). The UAV (including battery and propellers) weighs ~ 2.9 kg, has a maximum flight time of approximate 15 minutes, and is capable of flying at wind speeds up to 10 m/s. With zero wind, the UAV is capable of ascending with a speed up to 5 m/s, descending with a

speed up to 4 m/s and has a maximum horizontal speed of up to 22 m/s. When carrying the active AirCore as payload, the UAV system weighed ~ 4.1 kg and was able to make a ~ 12 minutes flight. The payload was attached to the bottom of the UAV using two 10 mm carbon fiber rods that were fixed to the UAV using zip-ties and duck-tape.





## 2.5 The analysis box

We constructed an analysis box to simplify the analysis of the air samples from the active AirCore, and to reduce the potential contamination of room air. A schematic of the analysis box is shown in Figure 2. Two female quick connectors for the reference and the fill gas are placed on the left side of the box. One of the two cylinders is selected via a solenoid valve

by the software of the CRDS analyzer. A needle valve and an excess flow path are situated between the solenoid and the rotary valve. The needle valve is used to restrict the total airflow that is set slightly larger than the flow rate through the CRDS analyzer, with the rest venting through the excess flow path. The rotary valve provides two positions, namely position A (Analysis) and position B (Bypass). The position is controlled via buttons outside the analysis box. Two 1/8" Swagelok bulkhead connectors are fixed to the middle of the box where the active AirCore is connected. On the right side of the

analysis box is the outlet, which is connected directly to the CRDS analyzer.

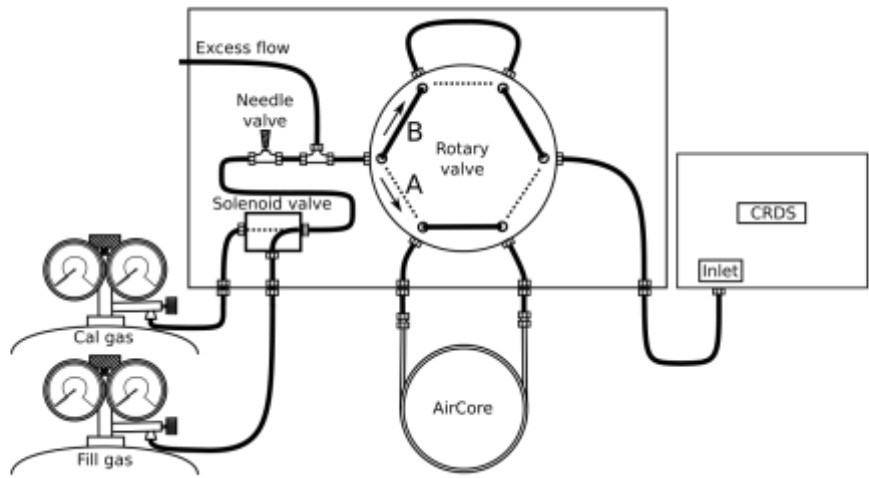

**Figure 2: a schematic of the analysis setup**

## 2.6 Data processing

One of the major advantages of the UAV-based active AirCore is that in contrast to a balloon-based AirCore, the UAV

normally lands next to the operator. This allows for immediate analysis of the air samples after landing and thus minimizes the spatial resolution degradation due to molecular diffusion of air samples in the tube. During flight, the CRDS analyzer is running a reference gas through a bypass path so that once the active AirCore is connected the analysis can begin immediately. Switching from bypass to analysis makes the reference gas 'chase' the active AirCore sample, while the analyzer drags the sample with a constant flow rate of 20.5 sccm. The sample is in fact analyzed in reverse, with the first

measured mole fractions linked to the landing of the UAV. The spiked CO mole fractions are seen towards the end of the analysis until finally the reference gas mole fractions are seen on the analyzer. This leads to a well-defined sample, seen as a





'plug' between the reference gas mole fraction values. Since the active AirCore is open on both ends, a small contamination from water vapor and ambient air is seen at the ends of each sample. Table 2 shows the mole fractions of $CO_2$, $CH_4$ and CO for the reference and fill gas, calibrated with respect to the WMO 2007 scale.

5 **Table 2: the calibrated mole fraction values of the reference and fill gas.**

|  | $CO_2$ [ppm] | $CH_4$ [ppb] | CO [ppb] |
|---|---|---|---|
| Reference gas | 390.8 ± 0.1 | 2010.9 ± 0.9 | 156 ± 1 |
| Fill gas | 411.4 ± 0.1 | 2027.7 ± 1.3 | 9376 ± 23 |

During the processing of the data the measured mole fraction values are corrected for water vapor as stated in section 2.3. A pre-determined calibration curve is applied to the measured dry mole fractions to correct for the small nonlinearity if there is any, and finally the mole fractions are corrected with a single bias between the measured and calibrated values of the

10 reference gas. Figure 3 shows the analysis of $CO_2$ (a), $CH_4$ (b), CO (c) and $H_2O$ (d) for the second flight made on September 13$^{th}$ 2016. The green and red dots indicate the start and the endpoint of the sample, respectively. The start point was selected as ¾ ways into the water vapor dip where the analysis goes from dried cylinder air to AirCore, while the endpoint was selected as the last point before the mole fractions goes above 2000 ppb CO, a little into the CO-spiked fill gas. The start and endpoints were consistently selected for all the flights.

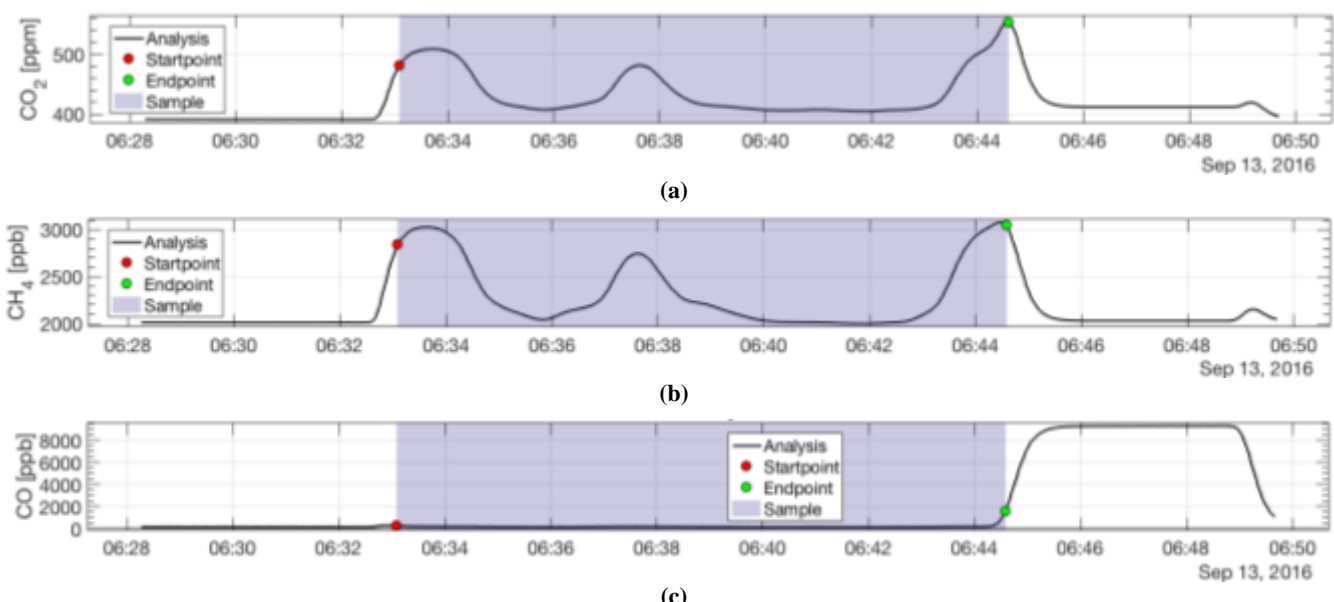





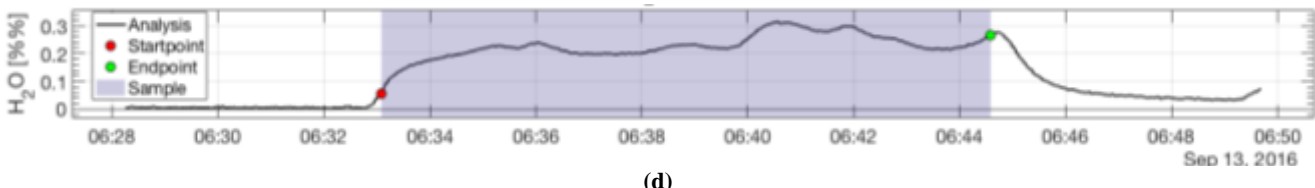

(d)

**Figure 3: the analysis of CO₂ (a), CH₄ (b), CO (c) and H₂O (d) for the second flight on September 13ᵗʰ 2017. The red and green dots indicate the start- and endpoint of the sample, respectively.**

The air entering the tube will quickly equilibrate with the mean active AirCore temperature. The pump creates a low pressure of ~ 380 hPa at the downstream end of the active AirCore, which is more than two times lower than the ambient upstream

pressure, forming a critical flow through the orifice. The length and the diameter of the active AirCore remain constant, and thus the only parameters that influence the sampling flow rate are the ambient pressure and the temperature of the AirCore and the orifice. Based on this and the ideal gas law, we estimate the number of moles of air ($\Delta n$) that flows into the active AirCore within a time step $\Delta t$ at any given time as the sum of the change of the number of moles of total air in the active AirCore and the number of moles of air flowing out of the AirCore:

$$\Delta n(t) = \frac{V}{R}\left(\frac{\Delta P(t)}{T(t)} - \frac{P(t)\Delta T(t)}{T^2(t)}\right) + \left(\frac{P(t) \cdot \Delta t \cdot f(t)}{RT(t)}\right) \tag{1}$$

Where $\Delta n$ is the number of moles of air sampled into the active AirCore, $P$ is the ambient pressure, $V$ is the total volume of the active AirCore, $R$ is the universal gas constant, $T$ the temperature of the active AirCore $t$ is the time and $f$ is the volumetric flow rate given by:

$$f(t) = C_d \cdot A \sqrt{\frac{R \cdot T}{M}} \tag{2}$$

where $C_d$ is the dimensionless discharge coefficient that can be empirically determined, $A$ is the area of the orifice, $R$ is the universal gas constant, $T$ is the temperature of the orifice in Kelvin, and $M$ is the molar mass of air in kg/mol.

During the analysis of the air samples by the CRDS analyzer, the flow is set at a constant rate. Therefore, the number of moles of air analyzed within a time step $\Delta t'$ at any given time $t'$ can be expressed as

$$\Delta n'(t) = \frac{P'f'(t)\Delta t}{RT'} \tag{3}$$

Where $f'$ is the analysis flow rate, $P'$ and $T'$ are the ambient pressure and temperature in the laboratory, respectively.

The number of moles of air samples that entered into the active AirCore during flight and the equal number of moles of air

samples analyzed by the CRDS analyzer are used to establish the link between the time it took to collect the sample and the time it took to analyze it.





Using equations (1), (2) and (3), an approximated flight-linked analysis time can be obtained, having effectively linked the number of moles going in to the sample with the analysis time. The measured mole fractions can then be directly linked to the time-series of the datalogger. Figure 4 shows the CRDS analyzer analysis with the original analysis time vs. the flight-linked analysis time.

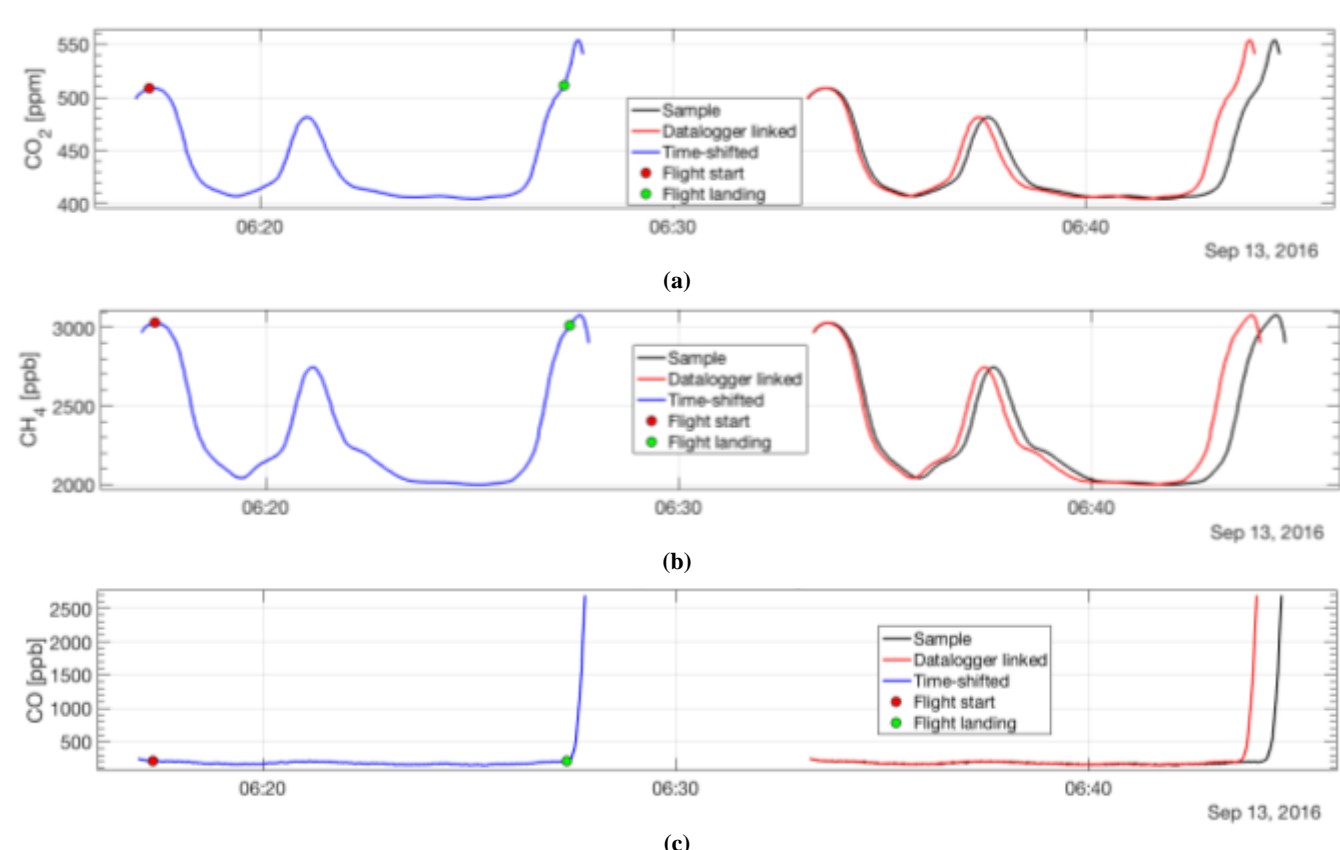

**Figure 4: the analysis of CO₂ (a), CH₄ (b), and CO (c) for the second flight on September 13th 2017 with its original analysis time, the datalogger-linked time series and the shifted datalogger-linked time series of the analysis. The red and the green dots represent the time when the drone took off and landed, respectively**

**2.7 Atmospheric station**

10  The atmospheric measurement station Lutjewad was established in the year 2000 by the Centre for Isotope Research (CIO) University of Groningen. The station is located 30 km from the city of Groningen and is easily accessible via roads, and is located on the northern coast of the Netherlands (6.3529º E, 53.4037º N, 1 m asl) situated directly behind the Wadden Sea dike. The coastal location of the station allows relatively clean marine background air to reach the sampling tower in contrast to more polluted air masses coming with the prevalent south-east to south-westerly winds. In analyzing wind direction data

15  for the years 2006 to 2014, it was found that the station received 16% of the time northerlies (315 - 45 degrees sector), 34% southerlies (135 - 225 degrees sector), 22% easterlies, and 28% westerlies. Hence, about half of the time the station receives





relatively polluted continental air masses. On the seaside, sporadically flooded salt marshes next to the dike pass into the Wadden Sea with its tidal flats. It stretches about six kilometers to the north where the island Schiermonnikoog marks the transition to the North Sea. The observatory itself is surrounded by insignificant shrubs and grass. The rural landscape to the south consists mainly of pasture and cropland with patches of forested land. The livestock in the area is dominated by dairy

cows and sheep. The nearest large town is the city of Groningen (200,000 inhabitants) at a distance of about 30 km in the ESE direction. The annual frequency of ESE winds, which could carry pollution from the city directly, is usually less than 1%.

   $CO_2$ and $CH_4$ were continuously monitored at 60 m via humid air analysis from a Picarro CRDS system model 2301, while measurements of $CO_2$, $CH_4$, and CO at 7 m was similarly measured using a Picarro CRDS system model 2401. The Picarro

CRDS measurements at the 7 m inlet were started a week prior to this campaign. The atmospheric station maintains continuous temperature, relative humidity and atmospheric pressure measurements at 7 m, 40 m and 60 m. At 7 m and 40 m, the wind speed is also measured, and at 60 m, the wind speed and wind direction. However, during the day of this study the wind speed and wind direction measurements at 60 m malfunctioned, and were not recorded.

## 3 Results

### 3.1 Flight trajectories

All flights conducted for this study were performed on September 13th 2016. The first three flights aimed to obtain vertical profile measurements of $CO_2$, $CH_4$, and CO. Information regarding the flight duration, time between flights, take-off location, landing location and mean speeds can be found in table 3. The first flight happened at dawn (06:15am local time), right before sunrise. The UAV ascended up to 210 m and hovered at this altitude for 45 seconds before ascending up to

500 m. The UAV hovered at this altitude for 20 seconds before descending back down to the landing zone. During the second flight, the UAV ascended up to an altitude of 300 m, and upon reaching this altitude, immediately started its descent towards an altitude of 60 m. Once this altitude was reached, it ascended back up to 180 m before starting its final descent towards the landing zone. The third flight trajectory was similar to the first flight, ascending from the take-off zone up to 500 m at a steady pace before descending back down to the landing zone. The datalogger malfunctioned during this flight,

causing the micro SD card to appear empty upon retrieval. This lead to no stored temperature, relative humidity, or pressure readings during this flight. For the processing of this flight, ambient pressure readings from the first flight were used to approximate similar altitude-pressures. The temperature profile from the first flight was used as the measured active AirCore temperature, but adjusted according to measured temperature profiles from the atmospheric station. The time series from the UAV flight log was used together with noted down times of when the pump was running to link with the analysis time. The

GPS coordinates and altitude was also obtained from the UAV log.

   The area between the northern dike and the coastal sea is covered with wetlands, and flight number four measured the $CO_2$ and $CH_4$ enhancement by flying from the dike to the sea. The take-off zone was located on the dike; having an elevation of





6.1 m. The UAV started at the take-off zone and ascended to an altitude of 22 m before flying horizontally over the wetlands towards the sea (north-western direction). The horizontal speed was averaging at 12 m/s for this leg of the flight. Once the UAV reached the sea, it descended to an altitude of 10 m and flew along the coastline (south-western direction) at an average speed of 4 m/s. Right before the UAV reached a critical battery level beyond the point of no return, it changed its direction and headed back towards the landing zone, cruising at an average speed of 5 m/s at an altitude of 10 m. At the landing site, the UAV hovered for 2 minutes before landing.

The fifth and final flight was a verification flight for the active AirCore system. The UAV hovered close to the 60 m tower inlet at the atmospheric station, sampling with the active AirCore while air at the 60 m inlet was pumped down to be analyzed by a CRDS analyzer in the ground station. Ascending to an altitude of 60 m, the UAV positioned itself next to the tower and hovered for 9 minutes before starting its descent towards the landing zone.

**Table 3: some of the common characteristics for the 5 different flights.**

|  | Flight #1 | Flight #2 | Flight #3 | Flight #4 | Flight #5 |
|---|---|---|---|---|---|
| Flight duration | 00:12:00 | 00:10:49 | 00:10:27 | 00:10:57 | 00:11:00 |
| Take-off time | 05:15:59 UTC | 06:17:00 UTC | 07:17:16 UTC | 08:21:48 UTC | 09:18:00 UTC |
| Landing time | 05:27:59 UTC | 06:27:49 UTC | 07:27:43 UTC | 08:32:51 UTC | 09:29:00 UTC |
| Time between flights | - | 00:49:00 | 00:49:27 | 00:54:05 | 00:45:09 |
| Take-off location | 6.3523 E, 53.4039 N, 2.3 m a.s.l | 6.3523 E, 53.4039 N, 2.3 m a.s.l | 6.3519 E, 53.4038 N 6.1 m a.s.l | 6.3518 E, 53.4041 N, 2.3 m a.s.l | 6.3525 E, 53.4039 N, 2.3 m a.s.l |
| Landing location | 6.3523 E, 53.4039 N, 2.3 m a.s.l | 6.3521 E, 53.4039 N, 2.3 m a.s.l | 6.3519 E, 53.4038 N, 6.1 m a.s.l | 6.3518 E, 53.4041 N, 2.3 m a.s.l | 6.3520 E, 53.4038 N, 2.3 m a.s.l |
| Mean ascending speed | 1.4 m/s | 1.5 m/s | 1.6 m/s | 0.2 m/s | 1.6 m/s |
| Mean descending speed | 1.8 m/s | 1.4 m/s | 1.7 m/s | 0.2 m/s | 0.9 m/s |
| Mean horizontal speed | 0.6 m/s | 0.3 m/s | 0.3 m/s | 5.4 m/s | 0.1 m/s |

### 3.2 Tower measurements

Figures 5 (a), (b) and (c) show the continuous measurements of $CO_2$, $CH_4$, and CO, respectively, on the full day of September 13th 2016. The 7 m inlet measurements are indicated with the black curves, while the 60 m inlet measurements are indicated by the blue curves. The vertical shaded lines represent the time interval of each of the five flights. As shown in figures 5 (a) and (b), the $CO_2$ and $CH_4$ mole fractions deviated strongly from each other at the times of the first and second



flights. During the third flight the 7 m and 60 m measurements were almost identical, indicating that the boundary layer below 60 m was well mixed. At the time of the third, fourth and fifth flight, a clear well-mixed boundary layer had formed. The third flight took place at 07:17:16 UTC, which was 09:17:16 local time.

**(a)**

**(b)**

**(c)**

**Figure 5: the continuous $CO_2$ (a), $CH_4$ (b), and CO (c) measurements from the atmospheric tower at 7 m (black) and 60 m (red). The highlighted areas indicate the timespan for each of the flights, approximately spaced one hour apart.**

As mentioned in section 2.7, the atmospheric station maintains continuous measurements of temperature, relative humidity and wind speed at 7 m, 40 m and 60 m. The time series during September 13[th] 2017 are shown in figure 6.

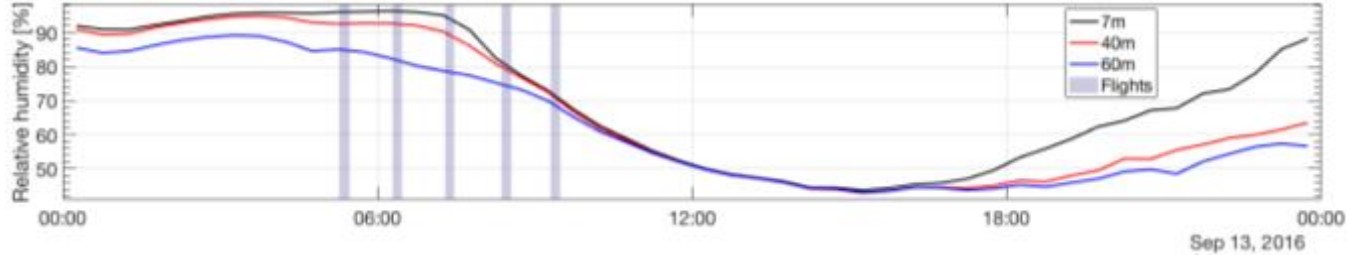





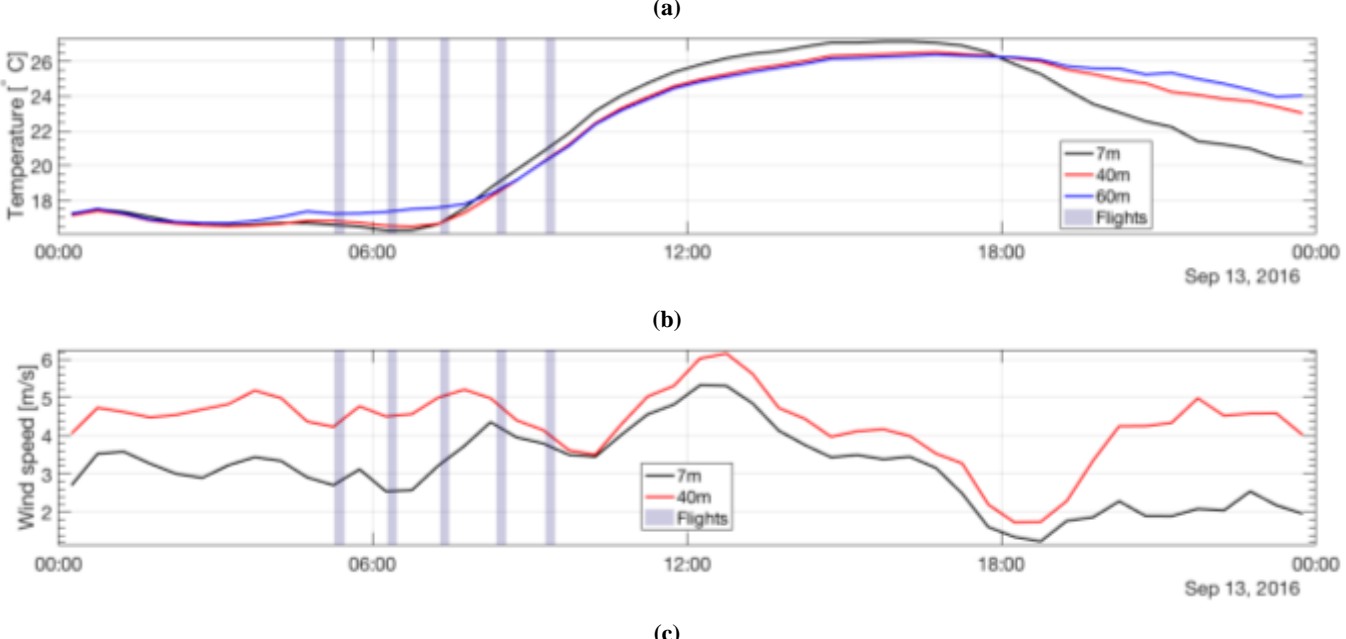

**Figure 6: the meteorological data measured at the atmospheric station during September 13th 2016. Figure 6 (a) shows the relative humidity, (b) the temperature and (c) the wind speed. The black curve indicates measurements at 7 m, the red curve at 40 m and the blue curve at 60 m. The highlighted areas indicate the times of the five flights**

### 3.3 The vertical profiles of $CO_2$, $CH_4$, and CO

5  Figures 7 (a), (b) and (c) show the measured mole fractions of $CO_2$, $CH_4$ and CO against altitude for the first three flights, respectively. Flight #1 is indicated by the red curve, the second flight the green curve and the third flight the blue curve. The solid lines indicate the ascending profiles, while the dotted lines indicate the descending profiles. The figures also show the measured tower mole fractions at 60 m and 7 m at the same time the drone was at these altitudes. Tower measurements for $CO_2$ are shown at 60 m in figure 6 (a), tower measurements for $CH_4$ at both 60 m and 7 m are shown in figure 7 (b) and

10  tower measurements of CO at 7 m are shown in figure 7 (c).





a)                                           b)                                           c)

**Figure 7: vertical profiles of a) CO$_2$, b) CH$_4$ and c) CO for flight nos. 1-3. Figures a) and b) include a dotted line indicating 60 m and shows measured trace gas mole fractions from the Lutjewad atmospheric station at this height. Figure b) include also a dotted line to indicate 7 m height and the corresponding CH$_4$ values obtained from the atmospheric station at this height. The square points represent the mole fractions measured at the time of the UAV ascent, and the circular points represent the mole fractions measured during the UAV descent. The color of the markers represents its respective flight. The CO mole fractions shown in figure 6 (c) has been averaged by every fifth data point. The ambient temperature and relative humidity is not shown due to the sensors only being placed inside the box, as discussed in section 2.1.**

The vertical CO$_2$ profiles seen in figure 7 (a) show how CO$_2$ mole fractions change throughout the morning hours. The vertical mixing of the boundary layer can be seen from the temporal change of CO$_2$ mole fractions that decrease at ground level from flight #1 to #2, and further from flight #2 to #3, coupled with a simultaneous growth of the CO$_2$ mole fractions between the flights at 60 m. This mole fraction growth at 60 m is also reflected in the CH$_4$ profiles shown in figure 7 (b). However, a decrease in CH$_4$ between flight #1 and #2 is not observed at ground level, which suggests an enhancement of methane has taken place between flight #1 and #2. The enhancement in CH$_4$ between flight #1 and #2 is confirmed by the observed CH$_4$ mole fractions at 7 m and 60 m from the Lutjewad tower (Figure 5 (b)). The enhancement is 470 ppb and 450 ppb for CH$_4$ at 7 m, and 60 m, respectively. These suggest a strong local surface source, likely from, ruminants and wetlands from the land surrounding the Lutjewad area. As seen in figure 5, a strong decoupling between 7 m and 40 m CO$_2$ and CH$_4$




until about 08:00 UTC+1 indicated a very shallow nocturnal boundary layer responsible for the high near-ground mole fractions associated with the local emission sources.

Above 200 m, the mole fractions of both $CO_2$ and $CH_4$ are nearly constant, with the exception of the $CO_2$ profile of flight #1. This suggests a stable boundary layer with a height of 200 m. However, we do not have a good explanation for the observed large variability of $CO_2$ seen in the descending profile of flight #1. Compared to $CO_2$ and $CH_4$, there is less variability in the mole fractions of CO, as seen in figure 7 (c). The enhancement in CO in the stable boundary layer relative to the CO aloft is seen for all the three profiles.

### 3.3.1 Validation against the atmospheric station measurements

Figures 7 (a), (b) and (c) also include the measured atmospheric station mole fractions of $CO_2$ and $CH_4$ at 60 m, and $CH_4$ at 7 m. The square markers indicate that the mole fractions were measured during the time the UAV was ascending, and the round markers indicate mole fractions measured during descent. The differences between the flight profiles and the tower measurements can be seen in table 4, where an average mole fraction from 50 – 70 m has been compared to the average mole fraction from the 60 m inlet during the same timeframe. Similarly, the average 7 m mole fractions within the given timeframe were compared to AirCore mole fractions between 0 – 20 m.

**Table 4: the differences between the measured active AirCore profiles and the trace gas mole fractions measured at the atmospheric station at 60 m and 7 m. An average mole fraction from the AirCore profile between 50 – 70 m is compared to an average mole fraction of the 60 m tower measurements within the same timeframe. Similarly, the average mole fraction from the AirCore profile between 0 – 20 m is compared to average a mole fraction of the 7 m tower measurements within the same time span.**

| | Trajectory | Horizontal distance between UAV and Tower | 50 – 70 m | | 0 – 20 m |
| --- | --- | --- | --- | --- | --- |
| | | | $CO_2$ difference [ppm] | $CH_4$ difference [ppb] | $CH_4$ difference [ppb] |
| Flight #1 | Up | 44 m | 12.869 ± 4.446 | 40.1 ± 28.8 | 19.5 ± 29.2 |
| | Down | | 6.162 ± 3.969 | -13.2 ± 33.7 | 57.7 ± 48.9 |
| Flight #2 | Up | 43 m | -7.930 ± 7.544 | -75.4 ± 79.4 | -46.8 ± 12.4 |
| | Mid | | -5.826 ± 3.896 | -87.1 ± 63.0 | - |
| | Down | | -0.076 ± 9.559 | -10.5 ± 30.3 | -37.3 ± 35.1 |
| Flight #3 | Up | 45 m | -0.223 ± 1.565 | -20.0 ± 25.6 | -1.4 ± 45.6 |
| | Down | | 0.146 ± 2.761 | 13.6 ± 19.3 | -20.0 ± 5.9 |



### 3.3.2 The variability of the flights

As seen in figure 7 (a), the behavior of the first flight with respect to the mole fractions of $CO_2$ did not follow expectations, nor did it have the same features as seen in the consecutive flights, and the features that are observed for $CO_2$ does also not

occur in the $CH_4$ or CO profiles. The correlation between $CO_2$ and $CH_4$ for flights 2 and 3 is strong, with $R^2$ values of 0.99 for both flights, while the correlation for the first flight yields an $R^2$ of 0.58. This low correlation could be due to $CO_2$ emissions from a nearby power plant. The Eemshaven coal power plant is located 34 km East of Lutjewad, and has a stack of 120 m. If the winds were not steady before sunrise, $CO_2$ emissions from the power plant may have dispersed to influence our flight profile, seen as the features in the figure 7 (a).

Both the descending and ascending mole fraction profiles during all the flights compare well with the continuous measurements of $CO_2$, $CH_4$, and CO at 60 m and 7 m, indicating that the features seen during the first flight's $CO_2$ profile is indeed real. The drop in the measured mole fractions at higher altitudes with each successive flight indicates that the boundary layer is transitioning from its nocturnal state to a mixed boundary layer. This is expected as the sun rises (Stull, 1988).

### 3.4 Methane enhancement from wetlands

Figures 8 (a) and (b) show the measured $CH_4$ and $CO_2$ enhancement relative to the background mole fractions measured at the atmospheric station during the fourth flight, respectively. The red color indicates a high enhancement of its respective trace gas, while the blue color indicates a low enhancement. The flight took place over the wetlands, north of the Wadden sea dike. The wind was from the southeast with a wind speed of $2.5 - 3.0$ m/s, which provided upwind measurements of $CO_2$

and $CH_4$ at the atmospheric station with respect to the flight. During the time of flight, the upwind measurements had a mean mole fraction of 2647 ppb of $CH_4$, and 460 ppm of $CO_2$. The $CH_4$ mole fractions were obtained from the 7 m inlet at the atmospheric tower, while the 60 m inlet provided the $CO_2$ mole fractions due to the low sampling frequency of $CO_2$ at 7 m. The mean altitude of the UAV during the flight was 10.4 m. The mean upwind mole fractions were subtracted from the mole fractions measured during the flight, providing the enhancement seen over the wetlands for each respective trace gas.

As seen in figure 8 (a) and (b), a clear hotspot for both $CO_2$ and $CH_4$ is seen towards the most northern part of the wetlands. The enhancement of $CO_2$ was at its peak 4.3 ppm over the background upwind measurements, and 85 ppb for $CH_4$, with a ratio $\Delta CH_4/\Delta CO_2$ of 19.8 ppb/ppm, which suggests that the emissions are from the local wetlands (Nara et al., 2014). The mean enhancement during the course of the flight was 1.2 ppm for $CO_2$ and 22.5 ppb for $CH_4$. The hotspot seen in figures 8 (a) and (b) were measured as the UAV was close to the coast. As mentioned previously, the wind was from the southeast,

further supporting that the measured hotspot originated from the wetlands. The southeastern wind direction made certain that the atmospheric tower measured the upwind mole fractions compared to the down-wind flight profile measurements. This means that the wetland was the sole additional contributing source to our profile.



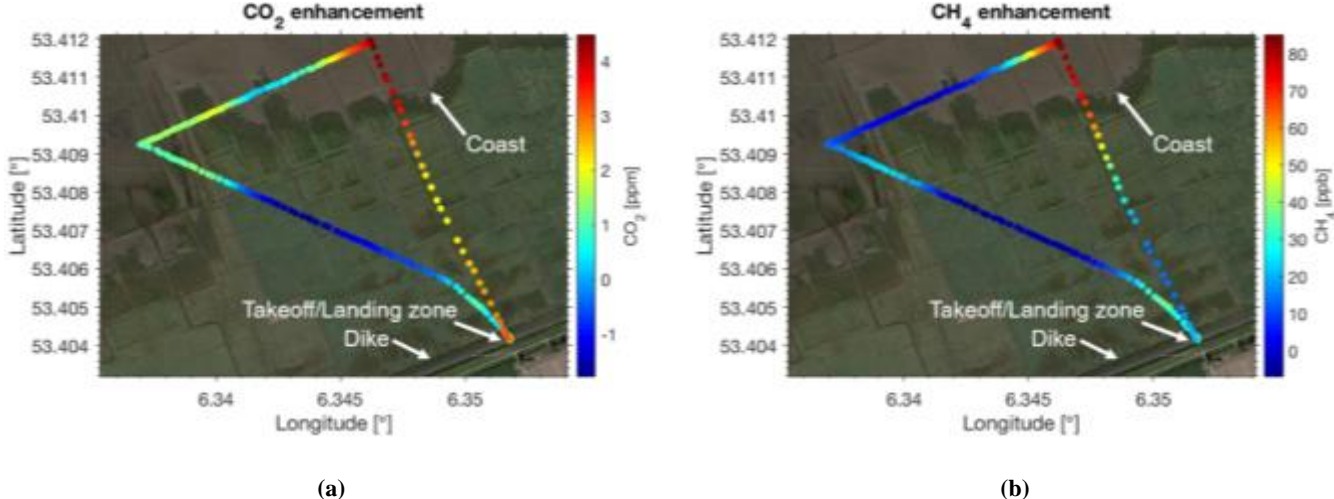

**(a)**                  **(b)**

**Figure 8: the measured mole fractions of $CH_4$ and $CO_2$ during the fourth flight. Take-off for the flight was on the dike, flying out towards the sea, doing a 90-degree turn and flying along the coast before heading back to the take-off spot. The red and blue colors indicate high and low mole fraction enhancement, respectively.**

### 3.5 Verification of the active AirCore

Figures 9 (a) and (b) show the measured $CO_2$ and $CH_4$ mole fractions from the fifth flight together with the measured mole fractions from the 60 m inlet at the time of flight. Figures 9 (c) and (d) show the correlation between the measured flight mole fractions and the 60 m inlet measurements for $CO_2$ and $CH_4$, respectively. Figures (e) and (f) show the mole fraction difference between the flight analysis and the 60 m inlet measurements for $CO_2$ and $CH_4$ respectively.

  As seen in figure 9 (a), the measured flight sample and the 60 m inlet measurements are in very good agreement throughout
the time of the flight. The first two minutes of the flight measure slightly higher $CO_2$ mole fractions than the continuous tower measurements, averaging 0.5 ppm above. An offset of the same size is also seen towards the end of the flight. Figure 9 (c) shows the difference throughout the flight, having a mean difference of $0.14 \pm 0.36$ ppm between the active AirCore and the 60 m tower inlet. Although the trend is similar, sharp peaks and troughs have been smoothed in the active AirCore compared to the tower measurements. There is a strong correlation between the active AirCore analysis and the 60 m tower
inlet measurements. This correlation is seen in figure 9 (e), and yields an $R^2$ of 0.97 for $CO_2$.

  As shown in figure 9 (b), the $CH_4$ analysis from the active AirCore and the 60 m inlet measurements follow the same trend. However, there is a consistent offset where the 60 m tower measurements measure higher mole fractions of $CH_4$. The difference throughout the flight is shown in figure 9 (f), having a mean difference of $-5.6 \pm 3.9$ ppb between the active AirCore and the 60 m tower inlet. The same smoothed curve as seen in figure 9 (a) is also seen in figure 9 (b). The sharp
peaks and troughs measured by the atmospheric station have been smoothed in the active AirCore. A strong correlation is seen between the $CH_4$ measurements of the active AirCore and the 60 m inlet analysis, and is shown in figure 9 (d). The $R^2$ is 0.95 for the $CH_4$ measurements.





Figure 9: the AirCore analysis of the fifth flight and the continuous tower measurements from 60 m. The plot shows the analysis profiles and the correlation between these two measurements from both $CO_2$ and $CH_4$. The differences in $CO_2$ and $CH_4$ between the two measurements are also shown.

## 3.6 Spatial resolution

The spatial resolution has four contributors, namely molecular diffusion, Taylor dispersion, smear effects of the analyzer and an innate uncertainty in the GPS measurements. Each contribution is discussed below.

### 3.6.1 Analyzer smearing effects

The cell of the analyzer also plays a role in the effective spatial resolution, in that is applies an additional smearing effect during the analysis. The sample flow rate through the CRDS analyzer is kept at a constant flow rate of 21.5 sccm. The volume of the analyzer cavity is 35 cc, but is maintained at 140 Torr (187 hPa) and 45 °C, which makes the effective cavity volume roughly 5.5 cc (at STP).





We use the response time (1/e exchange) to calculate the contribution of the smearing effect to the total spatial resolution, and determined it to be 15.3 seconds of the flight time. Considering the smearing effect alone, the spatial resolution of the active AirCore measurements is determined at 23.0 m with a mean ascent or descent speed of 1.5 m/s, or 38.3 m with a mean speed of 2.5 m/s.

**3.6.2 GPS uncertainties**

While the UAV is at a standstill, the uncertainty of the GPS is given as 0.5 m in the vertical direction and 2.5 m in the horizontal direction.

**3.6.3 Diffusion**

Molecular diffusion and Taylor dispersion that affects the profiled sample can be expressed with an effective diffusion coefficient, assuming that the flow is laminar through the active AirCore during sampling and analysis (Karion et al., 2010). The effective diffusion is expressed as

$$D_{eff} = D + \frac{a^2 \cdot \mathrm{v}^2}{48 \cdot D} \tag{4}$$

where $D$ is the molecular diffusivity of the different molecules in the gas (D is 0.16 cm$^2$ s$^{-1}$ for $CO_2$ and 0.23 cm$^2$ s$^{-1}$ for $CH_4$ (Massman, 1998)), $a$ is the inner radius of the active AirCore tubing and $\bar{v}$ is the average velocity of the air inside the active AirCore. The distance of diffusion $X_{RMS}$ is then given as

$$X_{RMS} = 2\sqrt{2 \cdot D_{eff} \cdot t} \tag{5}$$

where $t$ represents the storage time from the moment the UAV lands and the analysis is complete. The factor 2 in front of the square root comes from diffusion in both directions. The effective resolution in horizontal and vertical direction can then be expressed in terms of a fraction of distance travelled in space:

$$\Delta d_{diff} = \frac{X_{RMS}}{f} \cdot v' \tag{6}$$

where $\Delta d_{diff}$ is the effective resolution due to diffusion and dispersion, $f$ is the mass flowrate of the CRDS analyzer and $v'$ is the speed of the UAV. When the UAV is flying with an average speed of 1.5 m/s, the uncertainties range from 13.0 m to 16.5 m depending on the storage time. Storage time ranges from 5 to 20 minutes.

**3.6.4 Effective spatial resolution**

The effective spatial resolution can be calculated as a product of all the mentioned uncertainties, and is given by:





$$\Delta d = \sqrt{\Delta d_{diff}^2 + \Delta d_{smear}^2 + \Delta d_{GPS}^2} \tag{7}$$

Typical spatial resolutions are $\pm$ 44.0 to 47.0 m in the horizontal direction with a mean speed of 2.5 m/s, and $\pm$ 26.4 to 28.2 m in vertical direction having a mean speed of 1.5 m/s, with the Picarro CRDS smearing effect the major contributor.

## 4 Conclusions and discussion

In this paper, a UAV-based active AirCore was developed and was tested both in the laboratory and during flights. The laboratory test results show that the mean differences between the measurements of roof air by the active AirCore and a co-located CRDS analyzer are $0.04 \pm 0.21$ ppm, $0.58 \pm 0.67$ ppb and $0.86 \pm 27.37$ ppb for $CO_2$, $CH_4$, and CO, respectively. The direct comparison between the measurements of atmospheric air samples at 60 m from the active AirCore during flight and from the tower indicates a mean difference of $0.14 \pm 0.36$ ppm for $CO_2$ and $-5.6 \pm 3.9$ ppb for $CH_4$, respectively.

We demonstrate that the build-up of the boundary layer was clearly observed with three consecutive vertical profile measurements in the early morning hours. A clear enhancement in both $CO_2$ and $CH_4$ was captured during a low-altitude horizontal transect flight and was determined to be caused by emissions from the wetlands north of the Wadden sea dike.

The spatial resolution of the active AirCore samples is comprised of three factors: analyzer smearing effects, GPS uncertainties and diffusion, where the analyzer smear effect is the largest contributor. At typical speeds of 1.5 m/s for ascent and descent, and 2.5 m/s for horizontal flying, the effective spatial resolution is determined to be 26.4 to 28.2 m and 44.0 to 47.0 m, respectively, depending on the storage time. Due to the small amount of time between sampling and analysis (5-20 minutes), samples obtained using the active AirCore experience a low loss of sample resolution due to molecular diffusion. A modified CRDS analyzer with a reduced cavity pressure, e.g. 106 hPa or 53 hPa would greatly enhance the spatial resolution, since the response time of the CRDS analyzer would go down. Note that with a cavity pressure of 53 hPa, the spatial resolution is determined mainly by molecular diffusion, instead of the smearing in the analyzer.

The design of the volume and the length of the active AirCore, and the chosen sampling flow rate, provides up to 16 minutes of flight time. The range of the flights is largely determined by the performance of the UAV; however, the spatial resolution of the measurements is compromised by the speed of the flight.

The light weight of the active AirCore of 1.1 kg, its excellent preservation of the resolution of atmospheric air samples, and the mobility of a UAV lead to an effective sampling tool to measure greenhouse gases $CO_2$ and $CH_4$ mole fractions and a related tracer CO. This study shows the active AirCore's ability to capture both vertical and horizontal trace gas profiles with high precision and accuracy. The UAV-based active AirCore system opens up a wide variety of opportunities, including



measurements of GHG on a local scale with high resolution, quantifying $CH_4$ emissions from wetlands, landfills, other $CH_4$ hot spots and the quantification of $CO_2$ emissions from power plants.

*Acknowledgements*

*We would like to acknowledge Guido van der Werf for the loan of the VU CRDS analyzer, Henk Been, Dipayan Paul, Bert Kers and Joram Hooghiem for helping to wind up the coils of the Active AirCore, and Marcel de Vries for creating the data logger and the analysis box. Their help in the laboratory and preparation towards the campaign has been invaluable. We would also like to thank Toine Cornelissen and Vincent Lublink for providing the drone and for piloting and executing the flights.*

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
