# Peer review of "A UAV-based active AirCore system for measurements of greenhouse gases"

_Atmospheric Measurement Techniques, 2017_

## Editor Comment (EC1) · M. Leuenberger (Editor) · 16 Oct 2017

Dear authors

The quality of the graphs in the Discussion version should be improved.

---

## Author Comment (AC1) · 17 Oct 2017

Thank you for noticing this issue Markus Leuenberger. The low resolution on all the figures is strange indeed, and happened during the conversion from .docx to .pdf. I have uploaded a new version of the manuscript as a supplement that has much improved figure resolutions. I am sorry for the inconvenience.

Best regards, Truls.

Please also note the supplement to this comment:

[revised manuscript text omitted]

---

## Referee Comment (RC1) · Anonymous Referee #1 · 14 Nov 2017

General

This paper reports the development of active Aircore sampling from a UAV. This is potentially a very powerful new method of tackling a wide variety of important problems, such as measuring the flux from a gas leak, or quantifying the emissions from a large power station, cattle feedlot or wetland. The subject of the paper fully falls in the scope of AMT. The method is thoroughly explained, with good detail to allow replication by another team. The field testing is detailed and interesting. Thus the paper should be published.

That said, I have some specific points.

Specific Points.
Page 1. Introduction. This is very long winded, more suited to a thesis than a paper on measurement methodology. While I agree that some introduction is needed, maybe it would help readers if the wide ranging discussion on page 1 and page 2 up to line 22 was shortened considerably. It would be better to use the introduction to explain why measurement in the boundary layer is so valuable for CO2 and methane, and also potentially for other species, for example in pollution events. This is done briefly on P3 L13 on, but could be expanded. Active Aircore is a very powerful concept – say why it will be important.

P5 L 18 and 20. The 'box' – maybe give the box a name? – "Aircore box". You have another box later – "Analysis box" – and it's best not to get them muddled.

P6 L11. Relative humidity measurement. This seems to be a problem. Can you suggest a way around it, perhaps by relocating the sensor? It is important to have RH while sampling is taking place.

P6 Fig 1. – maybe move this figure a bit earlier, say into P5? – it would have been helpful from the start of the description.

P6 L20 – say clearly the CRDS at the landing ground of the UAV. It's a nice advantage of the method.

P7 L13 – is there a way you can test with undried air?

P8 L14 – in contrast to a FREE flying balloon. A tethered balloon doesn't have this problem. There are various ways to sample up to 400m – UAV aircore as here, UAV pulling up tube, free balloon, tethered balloon with tube, tethered balloon with active aircore. This method is good, but most of the others (except free balloon) have advantages too.

P9 L12 – $\frac{3}{4}$ ways into the water vapour 'dip' ?rise?

P11 L10. Are there any plans to test this against measurements at a really high tower? Cabauw for example?

P12 L3 – maybe 'low' shrubs – every plant is significant! P12 L14 – wind malfunction: pity. Such things always happen, but knowing the wind might help in the interpretation of the results!

P12 L16 – the paper reports a single day's experiments. This is fair enough as the paper is mostly about the method, but it would be nice to have a second trial. Maybe by the time the review process is complete it may be possible to add results from a second set of flights?

P14 and earlier – all the figures in my print out are very fuzzy and hard to read. They look like low resolution screen grabs? Maybe it is my system but if possible could some attention be paid to making the figures sharp and clear? Fig 7 is especially fuzzy and hard to read.

P18 Section 3.3.2. Also P19 L17. This is really interesting and it is a pity the RH and 60m wind measurements are missing. Maybe the discussion could be extended? Well worth repeating the flights, and doing some back trajectory work.

Conclusion

This is an important report of a valuable new technique. The paper should be published with minor revisions.

---

## Referee Comment (RC2) · Anonymous Referee #2 · 3 Jan 2018

General comments

The paper reports on the development and on a field test of an UAV-based active AirCore system for measurements of greenhouse gases. The subject of the paper fits AMT perfectly; it describes a method that bears a lot of potential for future research on GHG fluxes on scales of 100's to 1000's of meters. It is an important contribution to this field of research and should thus be accepted for publication – but only if the comments below are adequately and fully addressed.

The presented tests, validation and field deployment are in my opinion sharp at the necessary minimum maturity for the paper to be accepted. The paper shows an interesting way forward, but fails to present a robust method/application for the time being. Only after additional work it will become clear for what applications and to what extent

an active AirCore can be used best. It is critical that this is clearly communicated in the Discussion and Conclusion parts of the paper.

I fully agree with Anonymous Referee #1 regarding the comment concerning Page 1/Introduction. Furthermore, writing style improvements are possible throughout the text; avoid using statements devoid of a clear meaning where they are not adding any information or the reader expects clear, often quantitative information. The resolution of the figures must be improved. Also, every figure must be readable also if printed on an A4 sheet of paper.

Specific comments and technical corrections

Note on Technical corrections: in some cases, I have marked a word or formatting only once, but make sure to apply the corrections throughout the text where relevant.

Page 1, Line 1 (1/1): the word "accurate" from the title is not backed up by the paper's content (see e.g. 1/24!) – acceptable title is:" A UAV-based active AirCore system for measurements of greenhouse gases"

1/10: in some places, you are describing too many detailed information for an abstract (e.g. tubing dimensions).

1/14: Replace "…sample atmospheric air in both vertical and horizontal planes." …spatially sample atmospheric air."

1/16: delete "small" in "a small KNF micropump"

1/18: what is "… shortly after landing…"? for example use at least "not more than xx min after".

1/18: $H_2O$ should not be stated here; you are not calibrating for it, your sampled air is dried, it is not discussed in the text.

1/28: AirCore is not a platform – there are several platforms that would have access to locations you measure, but the question is what data and at what resolution it could

collect (and at what operational costs) – rephrase.

2/24: be clearer on the "vertical distribution" – I presume you are referring to the total column (as opposed to tall towers providing profiles, but only up to some 100's of meters).

4/1: here some relevant references are missing (Khan et al. 2012, doi:10.3390/rs4051355 ; Kunz et al. 2017, doi: 10.5194/amt-2017-207; Watai et al. 2006, doi:10.1175/JTECH1866.1)

4/23: please define "lightweight"

4/29: be more explicit on the analyzer you have used

5/Table 1: is +- 1 g a meaningful information? Are the numbers after the comma for the vertical and horizontal resolution representative/funded in your calculations?

5/4: "inner diameter (ID)", not "ID (inner diameter)", same for 5/5

5/6: rather use "glue for ceramics"

5/11: unclear sentence – please rewrite. "in the laboratory." implies that somewhere else, at another altitude this is different – clarify.

5/13: since you start describing the setup here, you should already here give details on the used hardware (e.g. pressure sensor type/model)

6/2: delete "product"

6/16: fox -> box

6/Fig 1: you can drop a and an before schematic and image. The photo should be cropped nearer to the box.

6/21: not necessarily true for H2O – delete this sentence; the relevant information on the analyzer's precision is included in line 7/2

7/2: please clarify where you got the numbers for precision from.

7/4 and 7/6: how do you measure/monitor the flow rates?

7/8: There must be more details / figures related to laboratory tests in the paper as they are a corner stone for the validation of any method/measurement setup. You are testing a setup that later flies on an UAV, but have not done any tests where the pressure on the inlet side varies. How many experiments were there (several?)? How large was the variation of the measured trace gases? Also, the parameters measured during tests and flights (e.g. pressure) is nowhere shown – but should be.

7/27: quad copter -> quadcopter

7/31: how exactly was it attached/what was the position of the inlet? The position of the inlet is important as the sampled air is influenced by the rotors (also depending on the direction and speed of the movement – particularly at relatively low speeds as are 1.5-2.5 m/s that you discuss in connection with the resolution).

8/1: specify the hardware used (i.e., quick connects, rotary valve, solenoid valve,...).

8/3: What do you mean by "contamination of room air"?

8/17:"reference gas"? In Fig. 2 you have Cal and Fill gases, only – please clarify.

8/18: replace "chase" (with "push" or " force")

8/21: This leads to a well-defined sample between the two reference gas mole fraction values.

9/14: This cannot be correct – see GAW Report No. 229 (2016; on p. 6: " The current scales are (as of June 2016): WMO CO2 X2007, WMO CH4 X2004A, WMO CO X2014A,...") and edit accordingly.

9/Table 2: the WMO CO calibration range is currently 30-500 ppb; see also 9/14

9/8: please explain what you mean with "...to correct for the small nonlinearity if there

is any,...".

9/12: "$\frac{3}{4}$ ways" – why? (and it is a rise, not a dip). In general, explain the starting and ending point choices.

10/Fig.3: "H2O [%%]"?

11/9: a map shoving the station and enough surroundings to meaningfully support the description of the station/the field experiment described later in the text is missing here.

11/12 "situated directly behind" is unclear – how far is "directly"? (see previous comment)

11/13 unclear/wrong sentence - rephrase sentence.

12/3: "The observatory itself is surrounded by insignificant shrubs and grass." – what exactly are "insignificant shrubs"?

12/8: I guess you mean 60 m a.g.l.? Any references describing the Lutjewad station measurement setup that you could cite here?

12/18: replace "happened" with "took place"

12/19: instead of "right before sunrise", better state the exact time of sunrise on that day.

13/Table 3: mean speeds are misleading as there was also some hovering involved in some flights (see also 7/31).

14/Fig.5: zooming into area / time of interest (all graphs) and adding measurement points from Aircore would largely increase the usefulness of these plots.

15/Fig.6: same as for comment above .

15/4: Is the time lag due to the long inlet lines at the tower taken into account in your calculations?

16/Fig.7: The titles above the graphs are not necessary. What are the fine dots in 7.c? As the five flights were so few and different it is difficult to say anything concrete on the quality or interpretation of the flights (particularly on flight #1). I therefore strongly suggest avoiding highly speculative interpretation attempts (as in 18/6-9). Some retro trajectories might help (even if a trajectory does not explicitly give information of the fluxes), but that might be already beyond the scope of this paper. 7 a and b are so much zoomed out that we can only poorly evaluate the performance of AirCore vs. tower – Table 4 is more helpful – some discussion is needed on why flight #3 seems to be giving the "best fit" profile, judging from Table 4 (even if there was no data recorded on the SD card).

18/10: "Both the descending and ascending mole fraction profiles during all the flights compare well with the continuous measurements of $CO_2$, $CH_4$, and CO at 60 m and 7 m, indicating that the features seen during the first flight's $CO_2$ profile is indeed real.": I cannot agree with this statement – the unexplained features of flight #1 are above the 60 m level. The fact that the measurements agree somewhat (not well) at 60 and 7 m does not imply that one knows what happened above 60 m – please rephrase.

18/19: from where did you obtain the information on the wind speed and direction (as the instrument on the tower was not recording this information)? For which altitude are the 2.5 – 3.0 m/s?

18/21: with what std.dev. for the mean mole fractions? And, you should state at first mention of "mole fraction" in the text that you are referring to "dry air mole fractions" (c.f. GAW Report No. 229, p.2)

18/30-31: mentioned already in 18/19-20

19/Fig.8: using the rainbow scale is not recommended (see e.g. https://www.climate-lab-book.ac.uk/2016/why-rainbow-colour-scales-can-be-misleading/ , https://www.poynter.org/news/why-rainbow-colors-arent-best-option-data-visualizations,etc.)
19/4: Chapter 3.5 is actually an attempt of validating the active Aircore measurements and could thus be part of Chapter 3.3.1. It is important to note that this "verification" is informative, but that it has only limited informative value for active flights, where the position of the UAV is changing (rapidly).

20/Fig.9e: there seems to be a clear trend – any ideas how to explain it? Could there be a systematic bias introduced during data processing?

20/7: do you mean transport delay and time constant?

21/20: how different are the uncertainties for CO2 and CH4 (having in mind their different molecular diffusivity)?

22/29: "This study shows the active AirCore's ability to capture both vertical and horizontal trace gas profiles with high precision and accuracy." I strongly disagree with this statement. Unless you find a definition of high precision and accuracy that fits your results, this sentence should be deleted/rephrased.

23/25: only cite what is published at time of writing

---

## Author Comment (AC2) · 22 Feb 2018

We would like to thank the reviewers for their useful and detailed comments, and we are happy with the positive feedback from both reviewers. In the revised paper, we have addressed the important comments regarding the lack of laboratory test information, poor figure resolution, the apparent trend in the difference between active AirCore and tower $CO_2$ measurements seen in figure 7 (e), and the different spatial resolution due to diffusion of $CO_2$ and $CH_4$. We feel these have especially improved the scientific quality and technical details of the paper. In addition, we have addressed the many minor remarks that was pointed out by the reviewers. A point-by-point answer to all the remarks from reviewer #1 can be seen below, given in red text.

**Reviewer #1 comments**

Page 1.  Introduction. This is very long winded, more suited to a thesis than a paper on measurement methodology. While I agree that some introduction is needed, maybe it would help readers if the wide ranging discussion on page 1 and page 2 up to line 22 was shortened considerably. It would be better to use the introduction to explain why measurement in the boundary layer is so valuable for CO2 and methane, and also potentially for other species, for example in pollution events. This is done briefly on P3 L13 on, but could be expanded. Active AirCore is a very powerful concept – say why it will be important.

- We have deleted four paragraphs, P2L21 to P3L22, regarding satellite, FTS and aircraft measurements, cutting out a big portion of the introduction to reduce the lengthiness.
- We've added two more sentences expanding on the importance and versatility of the Active AirCore: "… The Active AirCore provides a powerful tool to fill the vertical gap of GHG measurements between the surface and the lowest altitude usually reachable by aircrafts. The flexibility and mobility of the system makes it possible to make GHG observations at locations where tall tower measurements are not readily available. …"
- We've also added a few more references in terms of other uses of UAVs in atmospheric sciences:
  Watai et al., 2006, "… investigation of temporal and spatial variations of atmospheric $CO_2$ using a unique $CO_2$ measurement device attached to a small UAV (Kite plane) … "
  Kunz et al., 2017, "… and a dedicated $CO_2$ analyzer, COmpact Carbon dioxide analyzer for Airborne Platforms (COCAP), capable of being flown onboard small UAVs …"
  Khan et al., 2012, "… a small atmospheric sensor measuring $CO_2$, $CH_4$ and $H_2O$ attached to a robotic helicopter …" .

P5 L 18 and 20. The 'box' – maybe give the box a name? – "AirCore box". You have another box later – "Analysis box" – and it's best not to get them muddled.
- This is now called the AirCore box

P6 L11.  Relative humidity measurement. This seems to be a problem. Can you suggest a way around it, perhaps by relocating the sensor? It is important to have RH while sampling is taking place. P6 Fig 1. – maybe move this figure a bit earlier, say into P5? – it would have been helpful from the start of the description.
- The relative humidity sensor has been relocated to underneath the AirCore box in a newer version of the system. We've added a sentence stating this "… This has been resolved in the latest version of the active AirCore system, where the relative humidity sensor is now placed underneath the AirCore box. …". Figure 1 has been moved up to page 5.

P6 L20 – say clearly the CRDS at the landing ground of the UAV. It's a nice advantage of the method.
- We've added a sentence stating the landing site of the UAV close to the CRDS analyzer.
  "… , situated close to the landing site of the UAV. …"

P7 L13 – is there a way you can test with undried air?
- Yes, it should be possible to test with undried air. However, our roof inlet in the lab, which is the one that was used during the laboratory testing, is partially dried. New tubing would have to be implemented.

P8 L14 – in contrast to a FREE flying balloon. A tethered balloon doesn't have this problem.
  There are various ways to sample up to 400m – UAV AirCore as here, UAV pulling up tube, free balloon, tethered balloon with tube, tethered balloon with active AirCore. This method is good, but most of the others (except free balloon) have advantages too.
- Added the word 'free' too not confuse with the other types you mentioned.

P9 L12 – 3 4 ways into the water vapour 'dip' ?rise?
- Thanks for pointing out the error. We've changed it to the 'water vapor increase'.

P11 L10. Are there any plans to test this against measurements at a really high tower? Cabauw for example?
- We certainly seek opportunities to further validate and improve our active AirCore system in terms of the accuracy of both mole fraction measurements and the position registration of the air samples, e.g. at a tall tower, or comparing with aircraft measurements, or with in situ measurements of CO2/CH4 on UAVs that are likely less accurate in mole fractions but are more accurate in the position registration of the measurements.

P12 L3 – maybe 'low' shrubs – every plant is significant! P12 L14 – wind malfunction: pity.
Such things always happen, but knowing the wind might help in the interpretation of the results!
- Changed the word 'insignificant' to 'low'

P12 L16 – the paper reports a single day's experiments. This is fair enough as the paper is
mostly about the method, but it would be nice to have a second trial. Maybe by the time the review process is
complete it may be possible to add results from a second set of flights?
- Indeed, it would be nice to have a second trial; however, instead of a second validation experiments, we have
later used the active AirCore system to quantify $CH_4$ emissions from a dairy farm and from a coal mining shaft.

P14 and earlier – all the figures in my print out are very fuzzy and hard to read. They look
like low resolution screen grabs? Maybe it is my system but if possible could some attention be paid to making
the figures sharp and clear? Fig 7 is especially fuzzy and hard to read.
- Yes, certainly. It was due to an issue with the conversion from word to PDF. The figures will be shown in high
resolution in the final version.

P18 Section 3.3.2. Also P19 L17. This is really interesting and it is a pity the RH and 60m
wind measurements are missing. Maybe the discussion could be extended? Well worth repeating the flights,
and doing some back trajectory work.
- We would like to repeat the flights once we have obtained our own drone licenses. We have performed hysplit
backward trajectories to confirm get an indication of whether the parcels of $CO_2$ could have come from the
Eemshaven power plant.

---

## Author Comment (AC3) · 22 Feb 2018

We would like to thank the reviewers for their useful and detailed comments, and we are happy with the positive feedback from both reviewers. In the revised paper, we have addressed the important comments regarding the lack of laboratory test information, poor figure resolution, the apparent trend in the difference between active AirCore and tower $CO_2$ measurements seen in figure 7 (e), and the different spatial resolution due to diffusion of $CO_2$ and $CH_4$. We feel these have especially improved the scientific quality and technical details of the paper. In addition, we have addressed the many minor remarks that was pointed out by the reviewers. A point-by-point answer to all the remarks from reviewer #2 can be seen below, given in red text.

**Reviewer #2 comments**

General comments

The paper reports on the development and on a field test of an UAV-based active AirCore system for measurements of greenhouse gases. The subject of the paper fits AMT perfectly; it describes a method that bears a lot of potential for future research on GHG fluxes on scales of 100's to 1000's of meters. It is an important contribution to this field of research and should thus be accepted for publication – but only if the comments below are adequately and fully addressed.

The presented tests, validation and field deployment are in my opinion sharp at the necessary minimum maturity for the paper to be accepted. The paper shows an interesting way forward, but fails to present a robust method/application for the time being. Only after additional work it will become clear for what applications and to what extent an active AirCore can be used best. It is critical that this is clearly communicated in the Discussion and Conclusion parts of the paper

- The usefulness of a UAV platform to quantify instantaneous CH4 fluxes from a landfill has been demonstrated by Allen et al., 2018. Following this manuscript, we have used our UAV active AirCore system to quantify CH4 emissions from a coal mining shaft in Poland, and from a dairy farm in the Netherlands. The results of these studies will be published in separate papers, and are beyond the scope of this paper. We added a sentence in the conclusions and discussion "The usefulness of a UAV platform to quantify instantaneous CH4 fluxes from a landfill has been demonstrated by Allen et al., 2018."

I fully agree with Anonymous Referee #1 regarding the comment concerning Page 1/Introduction. Furthermore, writing style improvements are possible throughout the text; avoid using statements devoid of a clear meaning where they are not adding any information or the reader expects clear, often quantitative information. The resolution of the figures must be improved. Also, every figure must be readable also if printed on an A4 sheet of paper.
- We have deleted four paragraphs, P2L21 to P3L22, regarding satellite, FTS and aircraft measurements, cutting out a big portion of the introduction to reduce the lengthiness.
- All figures are now high-resolution PNGs, PDFs or EPS.

Specific comments and technical corrections

Note on Technical corrections: in some cases, I have marked a word or formatting only once, but make sure to apply the corrections throughout the text where relevant.

Page 1, Line 1 (1/1): the word "accurate" from the title is not backed up by the paper's content (see e.g. 1/24!) – acceptable title is:" A UAV-based active AirCore system for measurements of greenhouse gases"
- Changed the title to the suggested one. We would like to point out that the comparison results (P1/L24) are also affected by the real atmospheric variability, which adds noise/mean differences for short-term comparisons in general, and the accuracy of our active AirCore measurements is likely better than that.

1/10: In some places, you are describing too many detailed information for an abstract (e.g. tubing dimensions).
- Removed the tube dimension like the, i.e. Swagelok type, wall thickness etc.

1/14: Replace "...sample atmospheric air in both vertical and horizontal planes."..."spatially sample atmospheric air."
- Replaced this sentence with "spatially sample atmospheric air".

1/16: Delete "small" in "a small KNF micropump"
- Deleted throughout the paper

1/18: What is "...shortly after landing..."? for example use at least "not more than xx min after".
- Replaced "shortly after landing" with "not more than 7 minutes after landing"

1/18: $H_2O$ should not be stated here; you are not calibrating for it, your sampled air is dried, it is not discussed in the text.
- Deleted $H_2O$. $H_2O$ is still measured, and used to correct for the dilution and the pressure-broadening effects to derive dry mole fractions of $CO_2$, $CH_4$ and CO, as stated in section 2.3 and 2.6.

1/28: AirCore is not a platform – there are several platforms that would have access to locations you measure, but

the question is what data and at what resolution it could collect (and at what operational costs) – rephrase.
- *Changed 'platforms' to 'techniques'.*

2/24: Be clearer on the "vertical distribution" – I presume you are referring to the total column (as opposed to tall towers providing profiles, but only up to some 100's of meters).
- *This whole section of the Introduction has been removed in an attempt to shorten the introduction.*

4/1: Here some relevant references are missing:
*Khan et al. 2012, doi:10.3390/rs4051355 ;*
*Kunz et al. 2017, doi: 10.5194/amt-2017-207;*
*Watai et al. 2006, doi:10.1175/JTECH1866.1.*
- *We added the above mentioned references to the introduction.*
*P3L2-L3: "… investigation of temporal and spatial variations of atmospheric $CO_2$ using a unique $CO_2$ measurement device attached to a small UAV (Kite plane) (Watai et al., 2006), …"*
*P3L6-L7: "… , a small atmospheric sensor measuring $CO_2$, $CH_4$ and $H_2O$ attached to a robotic helicopter (Khan et al., 2012), …"*
*P3L8-L10: "… , and a dedicated $CO_2$ analyzer, COmpact Carbon dioxide analyzer for Airborne Platforms (COCAP), capable of being flown onboard small UAVs (Kunz et al., 2017). …"*

4/23: Please define "lightweight".
- *Added a parenthesis after "...lightweight UAV..." to specify: "... (total weight below 4 kg) ..."*

4/29: Be more explicit on the analyzer you have used.
- *Added a parenthesis specifying the analyzer: "... (CRDS, Picarro, Inc., CA, model G2401) ...".*

5/Table 1: is +- 1 g a meaningful information? Are the numbers after the comma for the vertical and horizontal resolution representative/funded in your calculations?
- *No, you are correct. The small uncertainty numbers were there to add all the information we had. We have removed these numbers.*

5/4: "inner diameter (ID)", not "ID (inner diameter)", same for 5/5
- *Switched the order*

5/6: Rather use "glue for ceramics".
- *Changed "ceramic glue" to "glue for ceramics".*

5/11: Unclear sentence – please rewrite. "in the laboratory." implies that somewhere else, at another altitude this is different – clarify.
- *Removed "in the laboratory". The vacuum the pump could provide was determined in laboratory, but you are correct in that the pressure is always monitored in the system, and does not change much from this value.*

5/13: Since you start describing the setup here, you should already here give details on the used hardware (e.g. pressure sensor type/model).
- *Added a parenthesis with the model number of the pressure sensor.*

6/2: Delete "product".
- *Deleted.*

6/16: fox -> box
- *Changed; "fox" is now "box".*

6/Fig 1: You can drop a and an before schematic and image. The photo should be cropped nearer to the box.
- *Done. The "a" and "an" are now gone, and the photo has been cropped closed to the box.*

6/21: Not necessarily true for $H_2O$ – delete this sentence; the relevant information on the analyzer's precision is included in line 7/2
- *Deleted*

7/2: Please clarify where you got the numbers for precision from.
- *They are based on cylinder measurements. We have added a sentence to express this: ''… , based on cylinder measurements before and after analysis of the AirCore. …".*

7/4 and 7/6: How do you measure/monitor the flow rates?
- *The flow rates were measured with an Alicat flowmeter located at the exhaust of the pump. The flowrate was noted at the beginning of the analysis and assumed constant throughout the analysis. It was monitored, but not logged. We have added a sentence on P6L13 – L15 stating this:*

"The flowrate was monitored using an Alicat MB-100SCCM-D/5M flowmeter located at the exhaust of the pump, and was noted down at the beginning of the analysis and assumed constant throughout the analysis of the AirCore."

7/8:     There must be more details / figures related to laboratory tests in the paper as they are a corner stone for the validation of any method/measurement setup. You are testing a setup that later flies on an UAV, but have not done any tests where the pressure on the inlet side varies. How many experiments were there (several?)? How large was the variation of the measured trace gases? Also, the parameters measured during tests and flights (e.g. pressure) is nowhere shown – but should be.

-     Three laboratory experiments were conducted to verify the consistency of the results. We have added this number to the text (P6L28). During the experiment, the datalogger tracked the inside pressure, the ambient pressure, and the temperature of the AirCore; all these parameters are essential for the processing of the data. The range of mole fractions during the experiments were between 394 to 417 ppm for $CO_2$, 2009 to 2120 ppb for $CH_4$ and 118 to 1657 ppb for CO. We have included the information regarding the range of mole fractions and which parameters were measured from P6L32 - P7L2:
"Figure 3 shows the time series of one of the experiments, The mole fractions during the three tests ranged from 394 to 417 ppm for $CO_2$, 2009 to 2120 ppb for $CH_4$ and 118 to 1657 ppb for CO. During the roof air tests, the datalogger tracked the inside pressure, outside pressure, and the temperature of the AirCore, which are the essential parameters that goes into the processing. From figure 3 (a) and (c), a small time lag between the AirCore measurement and the direct measurement can be seen. This is believed to be due to water vapor effects, as the air was not fully dried. "
We have also added a figure describing the laboratory setup (figure 2), and we have also added a figure showing the results of one of the three laboratory tests ($CO_2$, $CH_4$, and CO – Figure 3). a figure showing the results of one of the three laboratory tests.

7/27:    quad copter -> quadcopter
-     Changed "quad copter" to "quadcopter"

7/31:    How exactly was it attached/what was the position of the inlet? The position of the inlet is important as the sampled air is influenced by the rotors (also depending on the direction and speed of the movement – particularly at relatively low speeds as are 1.5-2.5 m/s that you discuss in connection with the resolution).
-     The inlet was facing downwards, sticking out from the bottom of the AirCore box. Added a sentence "..., so that the inlet was facing downwards towards the ground, ...", indicating the inlets orientation.

8/1:     Specify the hardware used (i.e., quick connects, rotary valve, solenoid valve,: : :).
-     Added manufacturer and model name to the individual parts.

8/3:     What do you mean by "contamination of room air"?
-     Changed from "… contamination of room air …" to "... reduce the potential contamination of the sample from non-sampled air...".

8/17:    "reference gas"? In Fig. 2 you have Cal and Fill gases, only – please clarify.
-     True, this is indeed confusing. We have changed the figure-text to "Reference gas", and changed the text accordingly.

8/18:    Replace "chase" (with "push" or " force")
-     Changed to "push".

8/21:    This leads to a well-defined sample between the two reference gas mole fraction values.
-     Added: "... between the two cylinder gas mole fraction values...". We did not want to state "...two reference..", as this may be seen as confusing with respect to the previously mentioned "reference" and "fill" gas terms.

9/14:    This cannot be correct – see GAW Report No. 229 (2016; on p. 6: " The current scales are (as of June 2016): WMO CO2 X2007, WMO CH4 X2004A, WMO CO X2014A,: : :") and edit accordingly.
-     We have edited accordingly. The calibration scales used are X2007 for CO, and X2004A for CH4 and CO.

9/Table 2: The WMO CO calibration range is currently 30-500 ppb; see also 9/14
-     Yes, it is. The fill gas CO content is not meant as an accurate number, it is there to be a clear indication as to when the sample ends. The calibrated CO value of the fill gas has a large uncertainty.

9/8:     Please explain what you mean with ": : :to correct for the small nonlinearity if there is any,...".
-     Changed the sentence from: "… to correct for the small nonlinearity if there is any, … " to "… to correct for drift in the linear calibration curve, …".

9/12:    "3/4 ways" – why? (and it is a rise, not a dip). In general, explain the starting and ending point choices.
-     Changed from a "... dip ... " to " ... increase ... ", and added a sentence stating that the points were empirically determined from the fifth flight.

P10L15: "… These points were empirically determined from the fifth flight, where the maximum correlation between the active AirCore and the 60m continuous measurements was found. These points were consistently selected for all the flights. …".

10/Fig.3: "$H_2O$ [%%]"?
- A typo; fixed by removing one "%".

11/9: A map shoving the station and enough surroundings to meaningfully support the description of the station/the field experiment described later in the text is missing here.
- We have added a google maps image showing the atmospheric station and its surroundings.

11/12 "Situated directly behind" is unclear – how far is "directly"? (see previous comment)
- Changed " ... directly behind ... " with "... roughly 50 m behind ... "

11/13 Unclear/wrong sentence - rephrase sentence.
- Agreed, it was a bit unclear. Decided to remove the sentence.

12/3: "The observatory itself is surrounded by insignificant shrubs and grass." – what exactly are "insignificant shrubs"?
- Changed to "... small shrubs..."

12/8: I guess you mean 60 m a.g.l.? Any references describing the Lutjewad station measurement setup that you could cite here?
- Yes, been changed to "60 m a.g.l.". We have added a reference that describes the station measurements setup: van der Laan et al., 2009.

  van der Laan, S., Neubert, R. E. M., Meijer, H. A. J., and Simpson, W. R.: A single gas chromatograph for accurate atmospheric mixing ratio measurements of $CO_2$, $CH_4$, $N_2O$, $SF_6$ and CO, Atmospheric Measurement Techniques, 2, 549-559, doi: 10.5194/amt-2-549-2009, 2009.

12/18: Replace "happened" with "took place"
- Changed "happened" with "took place".

12/19: Instead of "right before sunrise", better state the exact time of sunrise on that day.
- Time of sunrise has now been stated.
  "… The sunrise occurred at 06:05 UTC. … "

13/Table 3: Mean speeds are misleading as there was also some hovering involved in some flights (see also 7/31).
- Mean speeds have been removed from Table 3.

14/Fig.5: Zooming into area / time of interest (all graphs) and adding measurement points from AirCore would largely increase the usefulness of these plots.
- Now zoomed in to focus on the times around the flights instead of the full day. Sample data has also been included in the figure, along with a caption description of the altitude range for each flight: "… The altitude covered during the flights were 485m, 301m, 478m, 23m, and hovering at 60m for flights #1, #2, #3, #4, and #5 respectively. … ".

15/Fig.6: Same as for comment above .
- Now zoomed in to focus on the times around the flights instead of the full day

15/4: Is the time lag due to the long inlet lines at the tower taken into account in your calculations?
- Yes, the time lag has been accounted for.

16/Fig.7: The titles above the graphs are not necessary. What are the fine dots in 7.c? As the five flights were so few and different it is difficult to say anything concrete on the quality or interpretation of the flights (particularly on flight #1). I therefore strongly suggest avoiding highly speculative interpretation attempts (as in 18/6-9). Some retro trajectories might help (even if a trajectory does not explicitly give information of the fluxes), but that might be already beyond the scope of this paper. 7 a and b are so much zoomed out that we can only poorly evaluate the performance of AirCore vs.tower – Table 4 is more helpful – some discussion is needed on why flight #3 seems to be giving the "best fit" profile, judging from Table 4 (even if there was no data recorded on the SD card).
- Titles have been removed.
  The fine dots in Figure 7 (c) represent the CO data points with a time resolution of 3. The lines are drawn with a 5 data point average, as stated in the figure description. We've added a sentence in the figure text stating that each dot represents a data point:
  "… , with each dot representing a data point with a time resolution of 3 seconds. …". Added a possible explanation as to why the fit seems to be better for the third flight: "… From table 4, it is seen that the best fit

between data and atmospheric tower data occurred during the third flight. A possible explanation for this could be the smaller variability of mole fractions within the boundary layer. …".
We decide to keep the sentence that $CO_2$ may have been originating from Eemshaven  (east of Lutjewad), because this is the most likely interpretation we have based on available information. However, we have added more information to the matter, by including an additional sentence: "… Hysplit backward trajectories show that the winds emanated from the south-east during the time of the campaign.  …".

18/10:  "Both the descending and ascending mole fraction profiles during all the flights compare well with the continuous measurements of $CO_2$, $CH_4$, and CO at 60 m and 7 m, indicating that the features seen during the first flight's $CO_2$ profile is indeed real.": I cannot agree with this statement – the unexplained features of flight #1 are above the 60 m level. The fact that the measurements agree somewhat (not well) at 60 and 7 m does not imply that one knows what happened above 60 m – please rephrase.
- We have removed the sentence " ... , indicating that the features seen during the first flight's $CO_2$ profile is indeed real. ... ".

18/19:  From where did you obtain the information on the wind speed and direction (as the instrument on the tower was not recording this information)? For which altitude are the 2.5 – 3.0 m/s?
- The wind speed was recorded at the tower, just not at 60m. As figure 6 shows, the wind speed was recorded at both 40m and 7m, however, the tower did not provide wind direction. The wind direction, and also speed, was obtained from a monitoring station of the Royal Netherlands Meteorological Institute (KNMI) that is located in Lauwersoog (Latitude, Longitude, Altitude: [6.200E, 53.413N, 2.9m]) .
  Link to KNMI data: http://projects.knmi.nl/klimatologie/uurgegevens/selectie.cgi

18/21:  With what std.dev. for the mean mole fractions? And, you should state at first mention of "mole fraction" in the text that you are referring to "dry air mole fractions" (c.f. GAW Report No. 229, p.2)
- Added the standard deviation as well.

18/30-31: Mentioned already in 18/19-20
- Deleted the sentences 30-32.

19/Fig.8: Using the rainbow scale is not recommended (see e.g.
  https://www.climate-lab-book.ac.uk/2016/why-rainbow-colour-scales-can-bemisleading/
  https://www.poynter.org/news/why-rainbow-colors-arent-best-optiondata-visualizations ,etc.)
- Changed from the color scale "jet" (Rainbow) to "hot" (Black-Red-Yellow-White).

19/4:  Chapter 3.5 is actually an attempt of validating the active AirCore measurements and could thus be part of Chapter 3.3.1. It is important to note that this "verification" is informative, but that it has only limited informative value for active flights, where the position of the UAV is changing (rapidly).
- Moved section 3.5 into chapter 3.3.1.

20/Fig.9e: There seems to be a clear trend – any ideas how to explain it? Could there be a systematic bias introduced during data processing?
- We believe the bias is due to a contamination from unwanted air at the beginning of the sample, likely a contamination of human breath that introduced a big spike of $CO_2$ to the end of the sample, and hence contaminates that end of the AirCore sample with a higher concentration of $CO_2$. On the other end of the sample, the reference gas carried a lower concentration of $CO_2$ than the sampled air, and has likely contaminated the other end of the sample by lowering the overall concentration. This leads to the trend that is seen in the figure 11 (e). We've attached two figures to illustrate what we mean; The first figure is the AirCore analysis, with the black curve being the raw data and the red area the cut AirCore sample. The lower figure shows the comparison of this cut AirCore sample (still the red curve) along with a longer time series of the 60m Picarro measurements.
- We have added a sentence to explain this: "… Figure 11 (e) also shows a slight downward trend in the difference. This can be explained by contamination of the AirCore sample at both ends, where the end has been contaminated by a high mole-fraction $CO_2$ spike in one end, likely due to human breath while disconnecting the AirCore and preparing for flight, and the other side by the reference gas, which held a lower concentration of $CO_2$ than the sampled air. …"

[Figure]

-

20/7: Do you mean transport delay and time constant?
- No, The analyzer smearing effect is an effect due to the mixing of the air samples in the cavity of the CRDS analyzer.

21/20: How different are the uncertainties for $CO_2$ and $CH_4$ (having in mind their different molecular diffusivity)?
- The diffusiveness is not the dominant uncertainty in the calculation, the analyzer smearing effect is. However, it is true that you mention that $CO_2$ and $CH_4$ has different molecular diffusivity, which will influence the final uncertainty. The diffusivity is larger for $CH_4$, which will lead to a larger uncertainty for $CH_4$ than for $CO_2$, and so we've added a sentence stating the difference between the two:
"… Due to the difference in molecular diffusion for $CO_2$ and $CH_4$, the spatial resolution differs between the GHGs. When the UAV is flying with an average speed of 1.5 m/s, the uncertainties range from 7.6 m to 15.2 m for $CO_2$ depending on the storage time, while for $CH_4$ the uncertainty ranges from 9.1 m to 18.2 m depending on the storage time. Storage time ranges from 10 to 40 minutes. … "
We have changed the numbers in the paper abstract and conclusion to state the ones for $CH_4$, seeing how they carry the lowest spatial resolution..

22/29: "This study shows the active AirCore's ability to capture both vertical and horizontal trace gas profiles with high precision and accuracy." I strongly disagree with this statement. Unless you find a definition of high precision and accuracy that fits your results, this sentence should be deleted/rephrased.
- Deleted "... with high precision and accuracy...".

23/25: Only cite what is published at time of writing.
- According to the AMT guidelines: *Works "submitted to", "in preparation", "in review", or only available as preprint should also be included in the reference list.*

---

## Editor Comment (EC2) · M. Leuenberger (Editor) · 24 Feb 2018

Dear authors

Please upload a revised manuscript with track-changes along with the author-response, in order to judge your implementations as outlined in your response to the reviewers comments.

With best regards

Markus Leuenberger

—————————————————

---

## Author Response (AR2)

Dear Editor,

Thank you for your comments. See the replies for each individual comment below.

Comments to the Author:
Dear authors

Thank you for your revision of the manuscript. It significantly improved and gives now more information about the new technology you developed. I have only a couple of minor points that you need to address before publication.

P6,l20:   *Change the following statement: "The large standard deviation in CO is due to a sharp spike of several hundred ppb during three experiments, as seen in figure 3 (c). Figure 3 shows the time series of one of the experiments, The mole fractions during the three tests ranged from 394 to 417 ppm for $CO_2$, 2009 to 2120 ppb for $CH_4$ and 118 to 1657 ppb for CO."*
to
*"Figure 3 shows the time series of one of the experiments, the mole fractions during the three tests ranged from 394 to 417 ppm for $CO_2$, 2009 to 2120 ppb for $CH_4$ and 118 to 1657 ppb for CO. The large standard deviation in CO is due to a sharp spike of several hundred ppb during three experiments, as seen in figure 3 (c)."*
   - Changed the sentence to the suggested sentence.

Figure 3: Explain the cause why the CH4 does not agree at around 14:39.
   - The little spike in methane around 14:39 is likely due to metal-to-metal friction, by touching the stainless steel tubing during the analysis. We have added a sentence on P7L4 stating this.
   *"... Figure 3 (b) also shows a small $CH_4$ spike around 14:39. This is likely due to metal-to-metal friction, generated by touching the stainless steel tubing during analysis.  ...".*

[Figure]

   -

P10, l4-5: *"Table 2 shows the mole fractions of CO2, CH4 and CO for the reference and fill gas, calibrated with respect to the WMO 2007, 2004A and 2004A scales for CO2, CH4 5 and CO respectively"*
change to
*"Table 2 shows the mole fractions for the reference and fill gas, calibrated with respect to the WMO 2007, 2004A and 2004A scales for CO2, CH4 5 and CO respectively."*
   - Changed the sentence to the suggested sentence.

P11, l11: AirCore, t is...
   - Added a comma after "AirCore".

P16, l1: change blue to red curves!!
   - Changed from "blue" to "red"

P16, l3-5: The flight data are not mentioned (blue lines), add them. Why are the values significantly below the red lines despite hovering at 60 meters? As I understand from Figure 10, the flight tracks transects 60 meters but does not hover there. Therefore, change the verb hover to transect (transecting) in the Figure legend of 8.
   - Added a mention of the blue real-data curves:
     "… , and the blue curves indicate the sampled mole fractions during each flight. …".
   - The only flight that hovers at 60m was the fifth flight, all the others, with the exception of flight #4, were vertical profiles that went from 0 up to 500 m. That is why the mole fraction goes down below the

7m and 60m curves; the altitude is much higher, and the concentration lower. As you can see from the last flight, the blue real-data curve overlaps with very nicely with the 60m data, where the UAV active AirCore was hovering close to the 60m inlet. In comparing the transecting at 60 m with the vertical profiles, figure 10 gives the best description, and figure 11 gives the comparison of the hovering active AirCore with the 60m tower.

- The figure text for figure 8 has been changed from:
  *"Figure 8: the continuous CO2 (a), CH4 (b), and CO (c) measurements from the atmospheric tower at 7 m (black) and 60 m (red). The highlighted areas indicate the timespan for each of the flights, approximately spaced one hour apart. The altitude covered during the flights were 485m, 301m, 478m, 23m, and hovering at 60m for flights #1, #2, #3, #4, and #5 respectively."*
  To
  *"Figure 8: the continuous CO2 (a), CH4 (b), and CO (c) measurements from the atmospheric tower at 7 m (black) and 60 m (red), along with the mole fractions measured with the active AirCore (blue).The highlighted areas indicate the timespan for each of the flights, approximately spaced one hour apart. The altitude covered during the flights were 485m, 301m, 478m, and 23m for flights #1, #2, #3, and #4 respectively, transecting both the 7m and 60m altitudes. Flight #5 hovered at 60m close to the 60m tower inlet."*

Figure 8, 9: I would keep the red line for 60 meters as for figure 8. Take the blue line for 40 meters instead.
- For figure 9, we've changed the color of the 60m line to red, and the 40m line to blue, to be consistent with the colors used in figure 8.

Figures: For any Figure, e.g. Figure 10. Vertical - dot after number and start with capital letter.
- All figure and table captions have been changed to the above mentioned format.

P22, l19: I would list them according to the following description numbering.
  3.5.3 add Talyor dispersion, i.e. Diffusion and Taylor dispersion
- The line:
  *"The spatial resolution has four contributors, namely molecular diffusion, Taylor dispersion, smear effects of the analyzer and an innate uncertainty in the GPS measurements. Each contribution is discussed below."*
  has been changed to
  *"The spatial resolution has four contributors, namely smear effects of the analyzer, molecular diffusion, Taylor dispersion, and an innate uncertainty in the GPS measurements. Each contribution is discussed below."*
  to be consistent with the subsections in the following section.
- Added " … and Taylor dispersion …" to the section title.

P24, l8: with the major contribution from the Picarro CRDS smearing effect.
- Changed to the above mentioned sentence.

P24, l9: are instead of is
- Changed "is" to "are".

P25,l11: lower case for methane
- Changed CH4 to $CH_4$.

[revised manuscript text omitted]